# Learning Action and Reasoning-Centric Image Editing from Videos and Simulations

**Benno Krojer**
Mila & McGill University
benno.krojer@mila.quebec

**Dheeraj Vattikonda**[*]
Mila & McGill University

**Luis Lara**[*]
Mila

**Varun Jampani**
Stability AI

**Eva Portelance**
Mila & HEC Montréal

**Christopher Pal**
Mila & Polytechnique Montréal
Canada CIFAR AI Chair
ServiceNow Research

**Siva Reddy**
Mila & McGill University
Facebook CIFAR AI Chair
ServiceNow Research

## Abstract

An image editing model should be able to perform diverse edits, ranging from object replacement, changing attributes or style, to performing actions or movement, which require many forms of reasoning. Current *general* instruction-guided editing models have significant shortcomings with action and reasoning-centric edits. Object, attribute or stylistic changes can be learned from visually static datasets. On the other hand, high-quality data for action and reasoning-centric edits is scarce and has to come from entirely different sources that cover e.g. physical dynamics, temporality and spatial reasoning. To this end, we meticulously curate the **AURORA** Dataset (**A**ction-**R**easoning-**O**bject-**A**ttribute), a collection of high-quality training data, human-annotated and curated from videos and simulation engines. We focus on a key aspect of quality training data: triplets (source image, prompt, target image) contain a single meaningful visual change described by the prompt, i.e., *truly minimal* changes between source and target images. To demonstrate the value of our dataset, we evaluate an **AURORA**-finetuned model on a new expert-curated benchmark (**AURORA**-BENCH) covering 8 diverse editing tasks. Our model significantly outperforms previous editing models as judged by human raters. For automatic evaluations, we find important flaws in previous metrics and caution their use for semantically hard editing tasks. Instead, we propose a new automatic metric that focuses on discriminative understanding. We hope that our efforts : (1) curating a quality training dataset and an evaluation benchmark, (2) developing critical evaluations, and (3) releasing a state-of-the-art model[1], will fuel further progress on *general* image editing.

## 1 Introduction

Image editing is a complex task, involving many different skills, from adding/removing objects, changing colors/textures/styles, to **"taking actions": moving objects, changing actor positions or even more complex interactions**. Tackling all of these requires fine-grained understanding of how visual scenes are composed as well as **reasoning** (e.g. spatial instructions or referring expressions). No current model can successfully do all of these edits, and most only perform localized changes involving object addition/removal or attribute edits, following the "inpainting paradigm" [Zhang et al., 2024, Xie et al., 2023]. Others have tried to address this issue by introducing more specialized

---

[1]All data and code: https://github.com/McGill-NLP/AURORA     * = equal contribution

38th Conference on Neural Information Processing Systems (NeurIPS 2024) Track on Datasets and Benchmarks.

Figure 1: Previous failures on editing skills such as action, movement and reasoning (measured in AURORA-BENCH) compared to improvements with AURORA on these more challenging actions.

| Input | AURORA (Ours) | MagicBrush (strongest baseline) |
|---|---|---|

**Object / Attribute / Global:** *Let the horse wear a hat / Make the desktop black / Make it a picnic*

**Action / Reasoning:** *Make her walk away from the table*

**Action / Reasoning:** *Make the shampoo bottle fall down and land horizontally*

**Action / Reasoning:** *Move the mug to the right of the table*

model architectures which handle different editing subtasks [Couairon et al., 2023, Zhang et al., 2023a]. However, **neither of these approaches includes edits requiring more holistic visual understanding of how humans and objects interact or how events unfold**, such as 'make the cook cut the apple in half' or 'make the dog jump in the air' (see Fig. 1). These more *action-centric* edits are severely understudied in the space of instruction-tuned image editing models [Brooks et al., 2023, Huang et al., 2024]; when they are considered, it is done in isolation, ignoring other image edit subtasks and rigorous semantic evaluation [Souček et al., 2023, Black et al., 2024]. In Sec. 2 we describe a typology of these edit types and how existing datasets currently fail to address them all.

As we argue in this paper, **a major reason for these limitations is the lack of high-quality data**. Finetuning data of object or attribute changes is simpler to acquire than other forms of edits, since inpainting setups directly leverage strong object and attribute abilities of txt2img models [Rombach et al., 2022] for paired-image data generation [Yildirim et al., 2023, Zhang et al., 2024]. However, solving the data scarcity for learning action and reasoning-centric edits is not as straightforward. We identify videos and simulation engines as the two most promising sources of data for these edit types. As we discuss in this paper, we find that previous models trained on "noisy" synthetic image pairs or video frames lead to poor editing abilities. Here, noisy refers to image pairs with changes not mentioned in the prompt, i.e. due to shortcomings of the automatic generation process or inherent properties of videos such as viewpoint changes and non-meaningful movement. Therefore, our main requirement of high-quality action and reasoning-centric edit examples is that they be *truly minimal*: Edited images which contain one or maximally two semantic changes described by the prompt, while all other aspects are kept exactly the same. From a diverse set of video sources and simulation engines, we curate the AURORA Dataset (**A**ction-**R**easoning-**O**bject-**A**ttribute). Via crowd-sourcing and curation we collect 130K truly-minimal examples from videos and 150K from simulation engines for instruction-tuned image editing. We describe our dataset and collection process in Sec. 3.

The few image-text-alignment metrics commonly used in image editing are based on visual similarity to a groundtruth and in reality turn out to mostly measure the ability to stay maximally faithful (i.e. copying) to the source image [Zhang et al., 2024, Fu et al., 2023]. Though faithfulness is an important first step to master, these metrics have almost no correlation with the model's ability to

Table 1: **AURORA** vs. comparable public editing datasets. See details on all data sets in Sec. 2 and Sec. 3.2, and Sec. 3 for defining *truly minimal change*. ✓ = skill is covered but to a lesser extent.

| Dataset | Semantic Quality ('Truly Minimal Change') | Skill Coverage | | | |
|---|---|---|---|---|---|
| | | Obj. / Attr. | Global | Action | Reasoning |
| **InstructPix2Pix** | Low | ✓/ ✓ | ✓ | X | X |
| **HQ-Edit** | Low | ✓/ ✓ | ✓ | X | X |
| **GenHowTo** | Low - Medium | X/ ✓ | X | ✓ | X |
| **MagicBrush** | High | ✓/ ✓ | ✓ | X | X |
| + AG-Edit (Ours) | High | ✓/ ✓ | X | ✓ | ✓ |
| + Something-Edit (Ours) | Medium - High | ✓/ ✓ | X | ✓ | ✓ |
| + Kubric-Edit (Ours) | High | ✓/ ✓ | X | ✓ | ✓ |
| **= AURORA (Ours)** | High | ✓/ ✓ | ✓ | ✓ | ✓ |

generate accurate edits, especially on action and reasoning-centric changes. Hence, in addition to the training data in **AURORA**, we introduce **AURORA**-BENCH(Sec. 4), a manually annotated benchmark covering 8 editing tasks on which we collect human judgement (Tab. 2). Inspired by work on image generation models as discriminators [Krojer et al., 2023, Li et al., 2023], we also describe a novel discriminative metric that assesses *understanding* and hallucination (Sec. 5.1). To demonstrate the efficacy and quality of **AURORA**, we present a state-of-the-art instruction-tuned image editing model, finetuned on **AURORA** and evaluated on **AURORA**-BENCH, which we compare to strong baselines in a set of experiments in Sec. 5.3.

**In summary our contributions are**: 1) The creation of **AURORA**, a new clean and varied set of image edit pairs for instruction-finetuning that encompasses more action-centric and reasoning-centric examples. 2) We present a **comprehensive benchmark** covering a variety of edit types; 3) We introduce a **novel more informative metric** beyond existing ones; 4) We provide a **state-of-the-art image editing model** based on **AURORA** with well-rounded image editing capabilities covering object-centric, action-centric, and reasoning-centric edit abilities.

## 2 Typology of image edits

There are many ways to visually change a given scene [Huang et al., 2024]. **We define and focus upon five broad types of changes:** *object/attribute-centric, global, action-centric, reasoning-centric*, and *viewpoint*. Some can overlap: an action might change the attribute of an object, or reasoning can play a role in any type. Object-centric changes correspond to changes made to a specific object such as replacing it with another one, changing its attributes like color or texture, resizing it, or removing it entirely. Global edits change the overall image such as the background, style or textures. Action-centric changes correspond to changes that occur as a result of executing an action: changes in configuration of the objects, state changes of objects (e.g. cutting an apple), or pose change. Reasoning-centric changes are broadly defined as anything requiring compositionality or symbolic understanding: spatial, resolving referrring expressions ("sitting person on the far left"), negation, etc. Finally, viewpoint edits correspond to moving an egocentric camera, zooming in/out, and are the only on we do not cover in **AURORA** as they add many additional challenges: First, in videos *in the wild*, they exacerbate the already numerous changes; second, excessive camera movement can unpredictably alter the entire scene, introducing noise. App. C shows examples for each type.

**Coverage in existing data:** We characterize four broad sources of image pairs and prompts, which influence how much certain edit types are covered in existing training data (see Tab. 1):

1. Combining existing text-to-image models and LLMs in pipelines to automatically generate similar **synthetic** images [Brooks et al., 2023, Hui et al., 2024, Zhang et al., 2023b];
2. Providing **humans** with an image editing tool on existing images, combined with in-painting [Zhang et al., 2024], or finding existing Photoshop edits on the web [Tan et al., 2019];
3. Selecting nearby **video** frames and captioning the change via human annotators or automatically [Souček et al., 2023, Black et al., 2024, Alzayer et al., 2024];
4. Using **simulation** engines to generate pairs by precisely controlling visual changes with templated language [Michel et al., 2023, Wang et al., 2023].

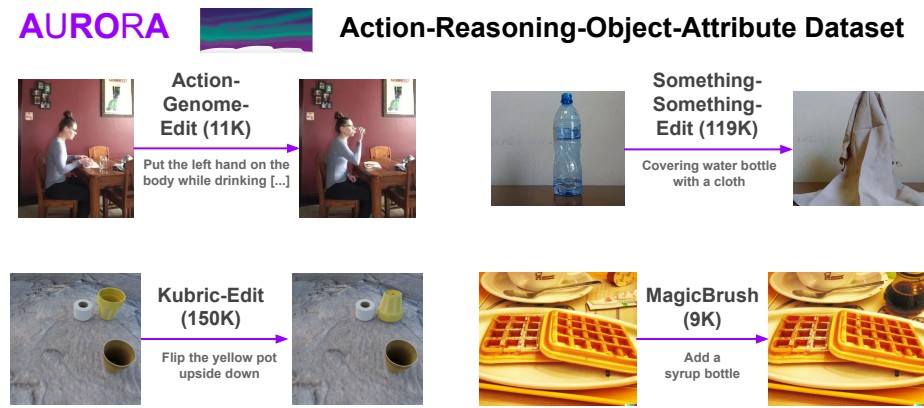

Figure 2: Our **AURORA** dataset covers action, reasoning and object-centric edits via 4 sub-datasets.

**InstructPix2Pix [Brooks et al., 2023]** introduced the first large-scale instruction-guided image editing dataset (313K) in a fully synthetic manner, combining GPT-3 (for prompt generation), Stable Diffusion [Rombach et al., 2022] and Prompt2Prompt (for increasing similarity of image pairs) [Hertz et al., 2022]. However, this scale comes at the cost of general data quality (see random samples in Fig. 28): Often Prompt2Prompt either fails to change anything or changes far more than asked for by the prompt, and in rare cases the prompt is non-sensical. This synthetic data is sufficient, though not optimal, for global and object-centric edits. However, action-centric and reasoning-centric edits either fail in execution or are not represented. Despite using more advanced models and pipelines, we find similar issues in **HQ-Edit [Hui et al., 2024]** under close inspection (see App. J). **MagicBrush [Zhang et al., 2024]** addresses some of InstructPix2Pix's shortcomings, mainly the lack of truly minimal edit pairs, with a rigorous crowdsourcing protocol where humans use the inpainting feature on the DALL-E 2 [Ramesh et al., 2022] interface. This methodology produces truly minimal image pairs for object-centric edits (see Tab. 3b). The inherent limitations of inpainting become apparent with certain attribute edits, and it is entirely unsuitable for action and reasoning-centric editing: Changing the color of a backpack via inpainting would also change its shape, size or texture. We did not find examples of action or reasoning-centric edits in MagicBrush (see App. J).

The landscape of actions and reasoning editing datasets is sparser: A relevant case is **GenHowTo [Souček et al., 2023]** which focuses on video frames that display actions and subsequent state changes in instructional videos. Their image pairs (and also captions) were chosen automatically, resulting in pairs that are not always minimally different due to excessive camera changes (App. J for random samples). We hypothesize that though GenHowTo may initially seem better at action-centric edits, like InstructPix2Pix it will tend to over-edit and not truly comprehend instructions due to training data quality issues. Reasoning-centric (spatial/geometric reasoning, referrring expressions, negation etc.) image pairs can be most directly created via simulation engines, with the hope of Sim2Real transfer. Simulations allows precise control over location, color and even orientation (rotation, flipping) of objects. Such reasoning is rarely covered in other sources: InstructPix2Pix and MagicBrush have almost no mentions of even the simplest spatial terms such as left" or "right". In the next section, we present **AURORA** which addresses some of the above shortcomings by using specific video sources and simulation engines to cover action and reasoning-centric edits in addition to existing quality object-centric editing data.

## 3 AURORA: A diverse and high quality image editing dataset

We present **AURORA**, a balanced dataset covering **A**ction **R**easoning, **O**bject and **A**ttribute edits, comprising a total of 289K training examples, see Fig. 2 and Tab. 1 for details and comparison to existing datasets. App. J provides 16 non-cherry picked training samples for all datasets.

### 3.1 Truly Minimal Visual Change

As surveyed in Sec. 2, many issues in previous datasets, even when they are large-scale and diverse, can be traced back to the lack of *truly minimal* image pairs. Beyond manually inspecting examples

in existing dataset, the lack of faithfulness wrt. the source image and prompt can be observed in generations of InstructPix2Pix and GenHowTo: In Fig. 3 models changed the background, color of the hydrant, etc. and in the case of GenHowTo, we tend to see "letter artifacts" from its training data (more examples in App. I). MagicBrush [Zhang et al., 2024] was able to produce much better object-centric edits simply by fine-tuning InstructPix2Pix on 8.8K truly minimal image pairs. To complement it, we create a novel (and larger) set of true minimal pairs for action-centric and reasoning-centric edits. Tab. 1 compares ours to existing datasets.

## 3.2 Creating quality data for action-centric and reasoning-centric edits

We use two types of sources to construct this new dataset: video and simulation engines.

**Videos** cover a wide range of action-centric edits as they are an abundant source of realistic and diverse state changes [Zellers et al., 2021, Miech et al., 2019, Niu et al., 2024]. However simply taking frames from any video data *in the wild* (e.g. YouTube) often leads to noisy data (see Sec. 5.4): the camera moves, too many things move at once, or the changes are simply not meaningful (i.e. easy to verbalize). Hence, we create *Action-Genome-Edit* and *Something-Something-Edit*, two new image editing datasets based on carefully selected video frame minimal pairs. Both datasets use frames from video datasets where humans were asked to do activities in or around the house through crowdsourcing which usually represents one action in isolation.

**For *Action-Genome-Edit***, we select frame pairs that had a CLIP cosine distance between 0.1 and 0.4, resulting in 15K pairs (thus filtering out many pairs with camera movement). Since no automatically generated instructions could reliably describe the changes, we tightly work with crowdworkers (App. D) to produce accurate edit instructions, with extensive quality screening and communication. Crucially, we ensured that workers discarded examples where a) there were too many or few changes, b) the changes were hard or lenghty to verbalize, or c) the camera moved (even if slightly) or the image quality was poor. After discarding 4K examples, the final Action-Genome-Edit dataset consists of 11K examples. **For *Something-Something-Edit*** we started from the original Something Something dataset [Goyal et al., 2017] which consists of 221K short clips where humans perform pre-defined basic actions such as "Attaching a string to a balloon", "Folding a cloth", "Lifting a book with a pencil on it", etc. Since the first and last frame of the short clips usually depict the start and end of the action, we selected them as our source and target images. We manually identify 10-15 categories of labels that don't lead to useful changes (e.g. "Pretending to..." where the person does not actually perform the action) and filter those out. The results is a set of 119K minimal frame pairs with high-quality simple edit instructions.

**Simulation engines** To perform action and reasoning-centric editing a model has to master spatial and relational reasoning, geometry, and simple movement. While videos provide some signal for learning these skills, a realistic simulation engine [Greff et al., 2022] offers full control over the arrangement and movement of objects. To teach this basic reasoning, we create *Kubric-Edit*.

*Kubric-Edit* contains 150K training examples which span three reasoning-centric edit skills – location changes, rotation changes, and count changes – and one object-centric edit skill – attribute changes. We build on top of Wang et al. [2023] who created 6K Kubric image pairs for contrastive image-text-matching, by defining more types of change and significantly extending the dataset. We manually filter and name more than 213 realistic objects from Google Scanned Objects, define templates for

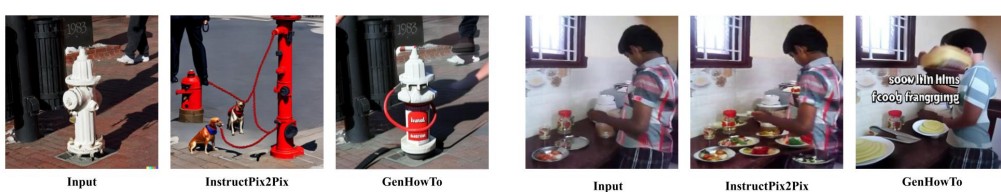

(a) *Add a leashed dog to the hydrant*          (b) *Show his hands on the plate arranging the food*

Figure 3: Common failure mode of previous models trained on noisy image pairs (e.g. InstructPix2Pix and GenHowTo): Their outputs are rarely faithful to the source image due to its noisy training pairs.

the edit instructions and ensure more truly minimal change. We cover actions such as *move, turn, swap, flip upside down, add/remove*, spatial configurations such as *left, right, up, down, rotation*, and attributes such as *size, shape or color*. More details are presented in App. E.

Thus, the **AURORA** dataset consists of four carefully selected sub-datasets coming from three sources: MagicBrush [Zhang et al., 2024] (humans equipped with an editing tool on MS-COCO), Action-Genome Edit and Something-Something-Edit (nearby video frames with collected high-quality human captions or filtered previous labels respectively), and Kubric-Edit (from the realistic simulation engine Kubric). In Sec. 5.3 we finetune InstructPix2Pix [Brooks et al., 2023] on our new dataset and thoroughly evaluate its performance across all types of edits. Note: Before collecting new data, we naturally tried to re-use existing datasets of visual change but found them inadequate for varying reasons described in App. M.1.

## 4  AURORA-BENCH: A holistic editing benchmark

To holistically assess the editing abilities defined in Sec. 2 (object/attribute, global, action, reasoning, excluding viewpoint), we manually **create a set of 400 image-edit-instruction pairs from 8 sources: AURORA-BENCH**. See Fig. 4a for an example of each one. We ensure that **AURORA**-BENCH allows studying out-of-distribution (OOD) transfer when a model is trained on **AURORA**, e.g. Sim2Real transfer from Kubric-Edit to real-world (spatial) reasoning or action edits outside of Action-Genome-Edit or Something-Something-Edit. Each of the 8 tasks contains 50 examples of image-prompt pairs that were either directly written by the authors or manually inspected for quality.

We cover object/attribute-centric edits with **MagicBrush** examples, action edits with **Action-Genome-Edit, Something-Something-Edit** and **Epic-Kitchen** [Damen et al., 2018] (OOD), reasoning-centric edits with **Kubric-Edit, CLEVR** [Park et al., 2019] (OOD) and **WhatsUp** [Kamath et al., 2023] (OOD); and **Emu-Global** covers global edits by sampling certain categories from [Sheynin et al., 2023]. MagicBrush, Action-Genome-Edit, Something-Something-Edit and Kubric-Edit are introduced in the previous Sec. 3.2. We manually select Epic Kitchen frames and write prompts to study OOD generalization of action understanding since the egocentric scenes and actions are quite different from the other two action-centric subtasks. To assess transfer from our Kubric simulation data, we leverage the real-world diagnostic data in WhatsUp for spatial reasoning, and OOD CLEVR images for testing spatial reasoning in addition to complex referring expressions.

## 5  Evaluation

We begin by introducing the metrics we use on **AURORA**-BENCH. Image-editing (and thus its evaluation) can be framed as a two step process: First, given an image-prompt pair, a model must understand how they relate to each other, for example by grounding phrases in the image. This is closely related to traditional vision-and-language understanding. Second, the model must perform the required edits by generating a new image, while being faithful to the original image. Previous work evaluates this second step – the final generation, which we also adopt as our primary judgement. However much insight can be gained from assessing understanding or *discrimination* abilities of editing models present in the first step. Our second evaluation proposes a new metric that tries to measure just that.

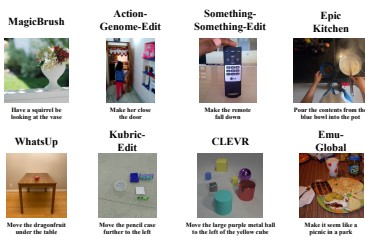 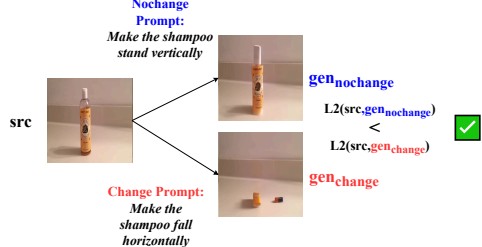

(a) **AROA-Bench**: Examples from our new expert-curated benchmark covering 4 editing skills (object-centric, action-centric, reasoning, global)

(b) **DiscEdit metric**: The left prompt describes the source: no change needed. The right prompt is a "normal edit", requiring a change.

## 5.1 Evaluation of final generations

**Existing metrics:** There currently exist a series of visual similarity metrics – L1, L2, CLIP-score, DINO-score – which are commonly used to evaluate the similarity of output edit compared to groundtruth images [Zhang et al., 2024, Fu et al., 2023]. The effectiveness of these metrics has not been formally justified for image editing, i.e. with human judgement correlations. We hypothesize that these metrics mostly reward models for copying information from the source image, rather than accurate editing. This hypothesis is confirmed using a trivial baseline, simply copying the source image as its output. This *copying* model outperforms all existing models on these metrics, see Tab. 3b. For instance, using human judgements of MagicBrush and **AURORA** outputs on MagicBrush test examples, we find a very weak correlation of 0.098 (using $CLIP - I$ score), also shown in [Ku et al., 2023]. Though faithfulness to the input is an important component of editing, so is actually *modifying* the image aligned with the prompt. These metrics also assume hard-to-obtain clean groundtruth images– without them meaningful automatic evaluation is even harder. Finally, since many automatic metrics rely on standard vision encoders (e.g. CLIP [Radford et al., 2021]) trained on static images (no video data) and known to be weak reasoners [Yuksekgonul et al., 2023], they are not suitable for our study. When we compute human correlation on WhatsUp examples (spatial reasoning), with clean groundtruth images, it drops to zero. In summary, these metrics might detect a model that struggles with faithfulness to the source image such as InstructPix2Pix (see Fig. 3), but are not helpful for comparing the semantic accuracy of stronger models.

**Human judgment of edited images:** With the insight that automatic visual similarity metrics are only a weak signal, we primarily rely on human judgment of model outputs on **AURORA**-BENCH: We ask humans to rate the absolute edit success (0=none, 50=partial, 100=full) as well as comparison (i.e. win-rates) between different models. We focus on the former in our main results as it allows us to compare task difficulty. We ensure that evaluators (we pick the best three evaluators from crowd-sourcing **AURORA**) pay most attention to the correct (semantic) interpretation of the edit prompts [2]. App. D.2 describes guidelines, compensation and extensive communication with crowdworkers.

## 5.2 DiscEdit: discriminative evaluation of image editing

We propose an additional automatic metric *DiscEdit* applicable to **AURORA**-BENCH examples. Unlike Sec. 5.1 above which considered the overall accuracy of generated edits, this evaluation serves as a diagnostic test for determining whether models truly understand how prompts relate to the input image, or can abstain from editing.

Inspired by text-to-image models repurposed as discriminators [Krojer et al., 2023, Li et al., 2023], models are given an image and two minimally different edit instructions $t_{nochange}$ and $t_{change}$. While $t_{nochange}$ requires *little to no change* to the source image, $t_{change}$ requires models to perform *significant changes*. An example of such a test pair is given in Fig. 4b. Thus, we expect the similarity between the first generated image $i_{nochange}$ and the source image $i_{src}$ to be higher than the similarity between the second generated image $i_{change}$ and the source image $i_{src}$, which we measure as a L2 distance in the latent space of the diffusion model (written $Enc(i)$):

$$\text{Score}_{\text{DiscEdit}} = \begin{cases} 1 & \text{if } \|\text{Enc}(i_{\text{src}}) - \text{Enc}(i_{\text{pos}})\|_2 < \|\text{Enc}(i_{\text{src}}) - \text{Enc}(i_{\text{neg}})\|_2 \\ 0 & \text{otherwise} \end{cases}$$

The intuition behind *DiscEdit* is that edits should be proportional to those described in the prompt – in other words, models should not change images more than required, nor should they produce fewer edits than requested in instructions. This metric therefore tests how much models are following or 'understanding' what instructions require. On the flip side, it also quantifies a form of hallucination: Changing things even when no change is required. Since it is not possible to find a "no-change" prompt $t_{nochange}$ for all kinds of prompts, we select a subset of source-prompt pairs $(i_{src}, t_{change})$ from *AROA-Bench* and manually define a no-change prompt $t_{nochange}$. App. H contains details on the data creation process and implementation of *DiscEdit*. A *DiscEdit* score of 0 or 1 is interpretable, and the metric does not require costly groundtruth target images. While it might seem far-fetched to expect the model to recognize when a scene does not need to be changed, we note that it is relevant in scenarios where image editing is a component of generative simulators [Yang et al., 2024].

---

[2]Inspections show high-quality ratings; inter-annotator agreement is a Fleiss-Kappa score of 0.626

Table 2: **Human Judgment** of semantic editing success on **AURORA**-BENCH tasks. Humans were asked to rate the edit success from none (0), partial (50) to full (100). Extended table in App. G. Overall score is "balanced": we average each skill first, and then take the average of those 4 numbers. Note: Our model was trained on more of the datasets than e.g. Magicbrush, so some of them are more IID for our model.

| Model | Obj./Attr. | Action, Human-Object-Interaction | | | Reasoning | | | Global | |
| | Magic Brush | Action-Genome | Something Something | Epic Kitchen | WhatsUp | Kubric | CLEVR | Emu-Global | Overall Score |
|---|---|---|---|---|---|---|---|---|---|
| GenHowTo | 18.0 | 8.0 | 8.7 | **17.7** | 4.3 | 0.7 | 2.0 | 11.3 | 10.8 |
| MGIE | 36.0 | 7.0 | 11.3 | 5.0 | 6.0 | 6.7 | 16.0 | 36.5 | 22.5 |
| InstructPix2Pix | 31.3 | 13.3 | 12.3 | 4.3 | 0.7 | 5.7 | 14.7 | 33.7 | 20.5 |
| MagicBrush | **61.7** | 16.3 | 17.0 | 12.0 | 3.0 | 9.3 | 22.0 | **42.3** | 32.6 |
| AURORA (Ours) | 60.5 | **35.6** | **31.8** | 14.2 | **27.3** | **59.6** | **46.1** | 33.0 | **41.3** |

Table 3: **Automatic Evaluation**: DiscEdit (left) and issues with existing automatic metrics (right)

(a) Discrimination performance with *DiscEdit* (comparing the two strongest models from Tab. 2). We show binary accuracy (50% random chance). Details: we average each example over 4 noise samples for the denoising process (App. H)

| Model | WhatsUp | Something | AG | Kubric | CLEVR |
|---|---|---|---|---|---|
| MagicBrush | 0.472 | 0.371 | 0.477 | 0.392 | 0.400 |
| AURORA | **0.565** | **0.548** | **0.583** | **0.592** | **0.450** |

(b) Automatic visual similarity metrics on MagicBrush test: Naively copying (!) the input is ranked better than the MagicBrush model for which this is IID (see Sec. 5.1).

| Model | L1↓ / L2↓ | DINO↑ | CLIP-I↑ |
|---|---|---|---|
| InstructPix | 0.112 / 0.037 | 0.746 | 0.8538 |
| MagicBrush | 0.072 / 0.025 | 0.865 | 0.915 |
| Naive Copy | **0.036 / 0.015** | **0.917** | **0.943** |

## 5.3 Results

Our baselines are InstructPix2Pix [Brooks et al., 2023], GenHowTo [Souček et al., 2023], MGIE [Fu et al., 2023] and MagicBrush [Zhang et al., 2024] [3]. See Sec. 2 for details on their training data. We train our own **AURORA** model with the InstructPix2Pix architecture on the **AURORA** dataset and mix all four sources such that each dataset is equally likely to be sampled during training, and take a checkpoint that was first pretrained on InstructPix2Pix and then MagicBrush (more details: App. F).

**Human Judgement** Tab. 2 summarizes our results on **AURORA**-BENCH evaluated via human judgement of successful adherence to the prompt and source image (0=none, 50=partial, 100=full). Most notably, **our model significantly improves on the challenging action and reasoning-centric edits**. However, action edits on complex real world images remain a challenge, while we see stronger numbers on "simpler" reasoning. At the same time, **AURORA** maintains strong performance on the diverse and well-established MagicBrush test set, leading to a high *Overall Score*. Finally, we observe generalization from training on simulation to CLEVR, and notably WhatsUp, a real-world spatial reasoning task.

**DiscEdit** Tab. 5 shows results with the *DiscEdit* metric on **AURORA**-BENCH examples. Discriminating between minmal prompts that either require a change or no change, proves to be a hard task: With **AURORA**, performance is slightly above random on most tasks except CLEVR. Performance of the MagicBrush model is even below random chance. We investigate this surprising result but could not find any pattern. Thus, we can only hypothesize that the wording of "no-change" prompts might be more unusual, and hence lead to hallucination behaviour (example outputs in App. I.2). We also find several encouraging qualitative results from our model (Fig. 4b).

## 5.4 Ablations and qualitative analysis

**Can we quantify Sim2Real transfer?** **AURORA** contains many simulated examples featuring spatial reasoning. To quantify the transfer to *spatial reasoning on real images*, we train a model on **AURORA** minus Kubric and manually rate the outputs on the WhatsUp examples from **AURORA**-BENCH: A win-rate of 46% for the full **AURORA** model, while without Kubric only a single win

---

[3]MGIE trains on InstructPix2Pix data; its main innovation is to enhance the original text encoder

(2%) is found [4]. We also find that training on truly minimal Kubric examples "stabilizes" training: Without it, the model hallucinates more un-needed changes and artifacts (e.g. adding people).

**EditDAAM:** We adopt DAAM (Diffusion Attentive Attribution Maps) [Tang et al., 2023] for qualitatively studying the attention maps of our editing model but study patterns across U-Net layers, grouping them into *Down*, *Middle* and *Upper* layers. We intuit that image *understanding* happens in earlier layers and the final *generation* in later layers. Since our model has seen more movement-based and spatial edits in training compared to MagicBrush, we hypothesize that this is reflected in its attention patterns. We illustrate these attention maps in Fig. 33 of the Appendix. Compared to MagicBrush, **AURORA** pays attention to a broader area starting in the middle layers of the U-Net, possibly since movement requires "scouting" the space where placing a new object is reasonable. In the upper layers it narrows down on precise object placement. Details in App. K.

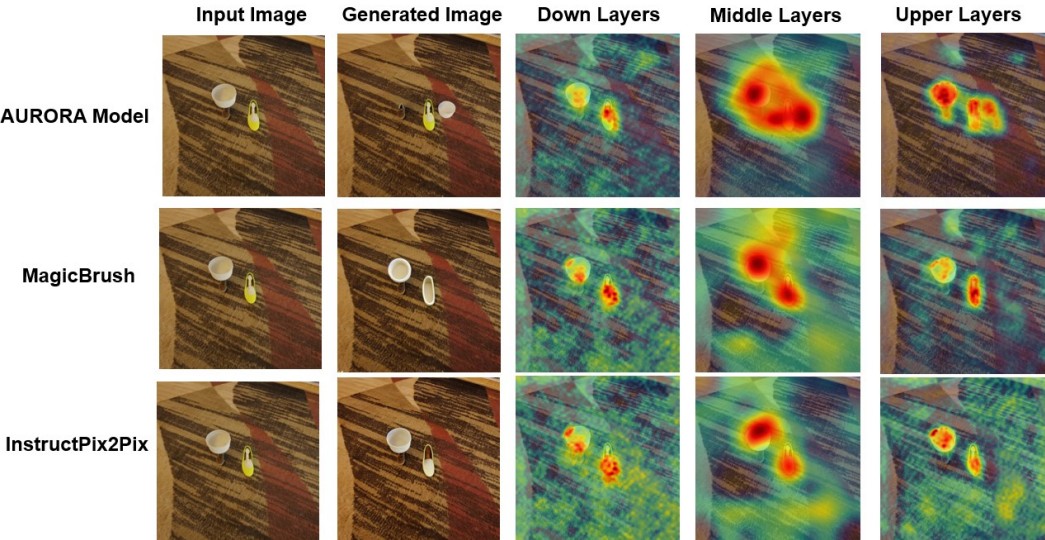

Figure 5: **Prompt:** *Put the white porcelain ramekin on the right hand of the brown shoe.* We show attention maps for three levels of U-Net layers: Down, Middle, Upper.

**Common verb and nouns in AURORA vs. MS-COCO:** We investigate if the distribution of verbs between broad generic VL captioning datasets and datasets tailored for action-editing differs significantly. A lot of edits have generic verbs like "move", which is nonetheless often still a complex task: "Move the cup closer to the plate" is a very different move than "Move the hand closer to their hair", where the exact action required is implicit in the scene/affordances/angles and not reflected in the textual "move". This is inherent to the editing task itself where captions tend to be shorter than traditional caption datasets, often with simpler verbs "make OBJECT ATTRIBUTE" (make the horse darker) or "replace/add OBJECT". So another interesting comparison is the distribution of verbs in traditional (object/attribute) editing vs. our focus on action editing. Finally, we note that a lot of complexity in our data comes from other linguistic constituents such as prepositions or adjectives/adverbs, e.g. "Move the hand slightly closer under the table with the finger pointing upward" where [slightly, closer, under, upward] are all interesting to understand but not verbs. To study the frequency differences to established caption datasets, we visualize the verb and noun distribution in MS-COCO, AURORA as well as the four subsets of AURORA. See App. L for detailed frequencies and figures. Overall, the verbs are less diverse but as described above a lot of the complexity comes from other textual or visual aspects. On top, the verbs are quite different to COCO and notably also quite different to more established object/attribute-centric editing such as MagicBrush. Also note that while "make" is a very frequent verb, it can often be accompanied with one of the other verbs like "make the person stand up".

---

[4]The rest are ties where both fully fail.

# 6 Conclusion and Future Work

Edits that require an understanding of real-world dynamics (e.g. actions) and reasoning are hard, especially compared to progress on more established editing subtasks. We hope that our contributions – from diverse high-quality training data with **AURORA**, to rigorous evaluation with **AURORA**-BENCH and a new state-of-the-art model– will pave the road for further progress on building *truly general* image editing models. Understanding how to improve action and reasoning-centric editing also relates to a more fundamental problem: world modelling, i.e. predicting the next observation after taking an action on the current one. This form of image editing can be seen as one-step controllable video generation, which in turn can be used as a **generative world-simulator** [Yang et al., 2024, Xiang et al., 2024, Zhou et al., 2024]. For example, both editing or video generation can "simulate" how a visual scene would change when a robot executes an action [Yang et al., 2024, Black et al., 2024]. Though, our results show that we are still far from achieving broad world models - see App. B.1 for a deeper discussion on limitations – our work is a step in that direction. It has the potential to not only enable better editing tools, but also to replace narrow rule-based simulators with generative ones for "limitless" interactive training data. It is an open question for future work whether one-step editing is the right paradigm or if generating the whole trajectory from source to target image (video generation) is needed to master the edits we study in this paper.

## Acknowledgements

This research was generously supported by Vanier Canada Graduate scholarship. We are also very grateful to Oscar Manas, Rabiul Awal, Vaibhav Adlakha and Marius Mosbach for their feedback and brainstorming ideas.

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

## Overview of Appendix

Our supplementary material contains the following, after the Checklist on the next page:

A **Dataset Release:** Everything from licensing to practical access

B **Broader Dataset Discussion** covers technical limitations and ethical issues

C **Examples of edit typology** illustrates the typology introduced in Sec. 2

D **Human annotation guidelines and details**

E **Data Curation Details** describes the technical details on how we collected or synthesized the data

F **Training Details**: hyperparameters etc.

G **Extended Tables** shows the main results with addtional confidence statistics

H **Details of DiscEdit** describes the implementation of our discriminative metric

I **Sample generations from our evaluation** shows random examples for both our evaluation setups

J **Random (=non-cherry picked) Samples from existing and contributed**

K **Details of EditDAAM**

L **Comparison of most common verbs and nouns in AURORA and established vision-and-language data (conducted for rebuttal)**

M **Behind the scenes** shows not just the final product (this paper) but also how we arrived here, what we discarded, and some personal reflections

N **Datasheet for Dataset "AURORA"** training datasets


## A Dataset Release

In this section we document the details of our dataset drawing from existing frameworks such as Datasheets for Datasets [Gebru et al., 2021].

1. **Data and code**: `https://github.com/McGill-NLP/AURORA`

2. We provide a **Datasheet** [Gebru et al., 2021] for our **AURORA** and **AURORA**-BENCH data as a Markdown file: `https://github.com/McGill-NLP/AURORA/blob/main/datasheet.md`, as well as at the end of our appendix: App. N.

3. **Hosting Plan**: As described in our instructions on the GitHub repository, we host our data on Zenodo and intend to put it on Huggingface soon after submission

4. **Licensing**: We release all our data (AURORA, AURORA-BENCH) under the **MIT License for easy access to other researchers**. The license allows users to share and adapt the dataset for any purpose, even commercially, as long as appropriate credit is given and any changes made are indicated. The datasets we build upon or directly include in our collection of data have the following licenses: MagicBrush (CC 4.0), Action Genome (MIT), Something Something (see `https://developer.qualcomm.com/software/ai-datasets/something-something` for their terms of use, more restrictive than MIT or CC 4.0).

5. **Consent & Privacy**: We work with Amazon Mechanical Turk crowdworkers who did not share any private information. Two of the datasets we build on top of, Action Genome [Ji et al., 2020] and Something Something [Goyal et al., 2017], asked humans to film videos at home doing daily activities. Action Genome builds on top of Charades [Sigurdsson et al., 2016], and we did not find any mention in their paper discussing privacy or consent: Workers were recruited on AMT. The situation is similar for Something Something but we would assume that workers were told and aware that their data is going into a public research dataset. This is also officially part of the agreement when becoming a worker on AMT.

## B    Broader Dataset Discussion

### B.1    Limitations

We acknowledge that these models are not yet mature to robustly perform the edits we study in this paper: Even in simpler setups such as Kubric, WhatsUp or CLEVR we still observe some failures, and on messy data where humans perform actions fully correct edits are rare, as reflected in the human ratings (Tab. 2. Even on the more established object-centric and global we still identify many failures, arguably more than in the more mainstream text-to-image generation.

**Specifically we identify the following failure modes during many manual :** Models pick up artifacts if something is overrepresented in AURORA: It might over-generate hands or people due to many such examples in the video-frame-based data. Similarly, the model sometimes falls back to textures and shapes from Kubric when common Kubric phrases (i.e. cups) are mentioned. While our model has learned spatial relations, it still struggles with "truly moving" an object: Sometimes the original objects is kept at its place while a new one is added elsewhere, resulting in too many objects. Often, the properties of the object (i.e. size or texture) also change, and can become more "Kubric-like" after being moved. Finally, while many Kubric edits were successfully performed on its IID test examples, *swapping the position* of two objects was not.

### B.2    Ethical and Societal Discussion

While we envision robotics and other planning tasks as an exciting new application of models that can perform action and reasoning-centric edits, there are several potential harms specific to the broader editing task. The main one is editing images in harmful or privacy-intrusing ways. These harmful edits are not in our training data, but the underlying Stable Diffusion model was trained on them and thus sometimes generates various harmful or biased images [Birhane et al., 2021, Luccioni et al., 2023]. We found the efforts of MagicBrush [Zhang et al., 2024] helpful, a dataset we include in our AURORA dataset, such as minimizing these problems in their collection as described in their appendix.

On the crowdsourcing side, we ensured fair pay and treatment with compensation far above the minimum wage in the US, see App. D. On top of pay, workers gave us feedback several times that they appreciated the feedback, respect for their work and detailed communication.

## C    Examples of edit typology

In this section, we illustrate our typology of edit skills from Sec. 2. For more examples for each dataset (not skill!) see App. J.

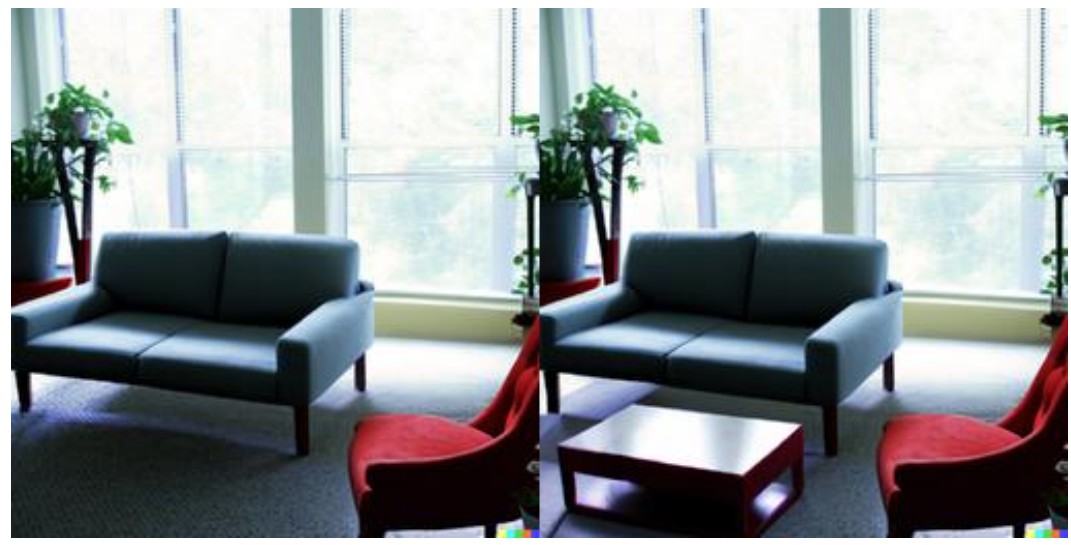

Figure 6: **Object-centric edit example**: *Can we have a wooden table?*

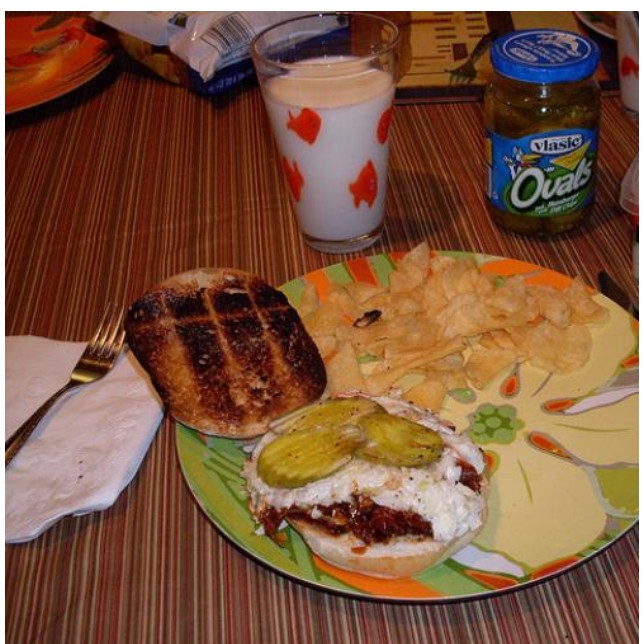

Figure 7: **Global edit example**: *Make it a picnic*

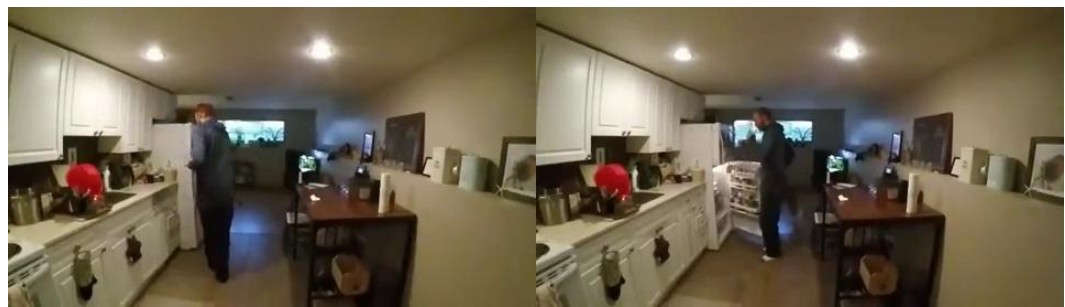

Figure 8: **Action-centric edit example**: *Make the man open the refrigerator*

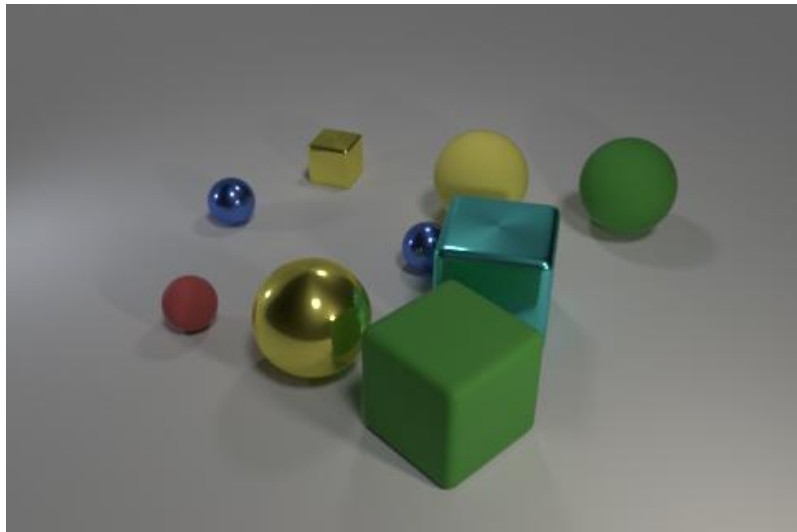

Figure 9: **Reasoning-centric edit example (spatial, referring expression)**: *Move the green sphere in front of the red small sphere*

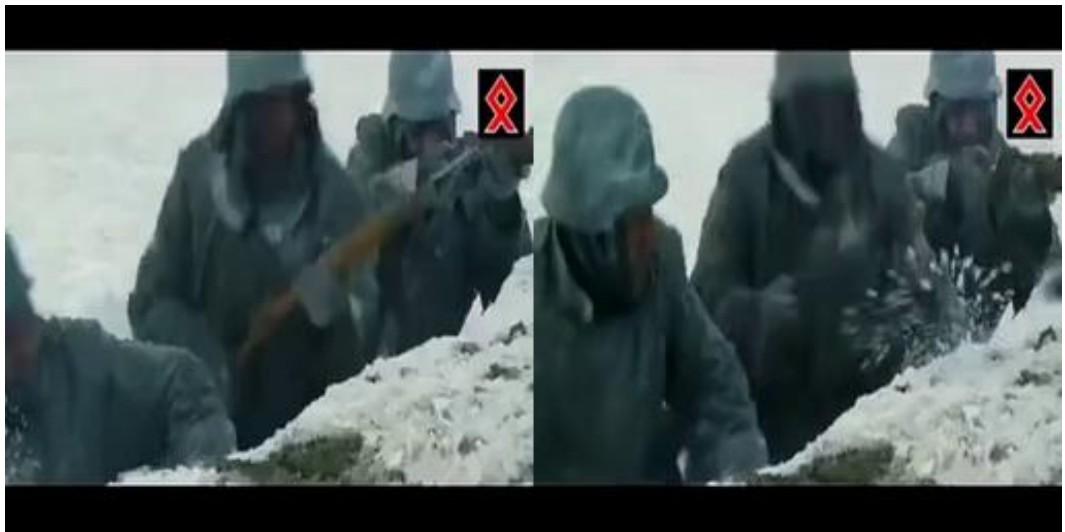

Figure 10: **Viewpoint edit example (from a dataset we later discarded, see App. M)**: *Move the camera left a bit to capture the first person on the left well*

## D   Human annotation guidelines and details

We work with a smaller group of expert annotators on AMT (who we individually contact via e-mail) after an initial screening during pilot runs. Seven workers first worked on describing changes between nearby video frames as edit prompts, and later on three of those workers for the human judgement. We found that (unsurprisingly) paying well and regular detailed communication led to the very clean results and resulted in, to put it directly, amazing feedback working with us as data collectors: We paid 0.22 USD for the captioning, and 0.20 USD for the human judgement. We estimated this to be significantly above 10 USD/hour, and probably closer to 15 USD/hour. Below we describe the instructions and communication with workers for each task.

## D.1 Crowdsourcing edit instructions

Our instructions on the AMT interface looks as follows:

**Instructions**

**You are given two nearby frames from a video and have to describe the change. Specifically, imagine you are trying to describe to an AI editing model how it has to modify the first image to reach the second one.**

All the instructions and examples are in the email. If you see this, you should have received it. Otherwise please contact me.

Remember the three steps to ask yourself before your write the caption:

1. Is there any (even small) camera movement? (Tip: check the four corners: bottom left, bottom right, top left, top left) --> DISCARD
2. Is it really blurry or dark etc and therefore hard to see anything? DISCARD
3. Is there a single well-defined change, i.e. it can be described in a single normal-length sentence? Write a change caption!

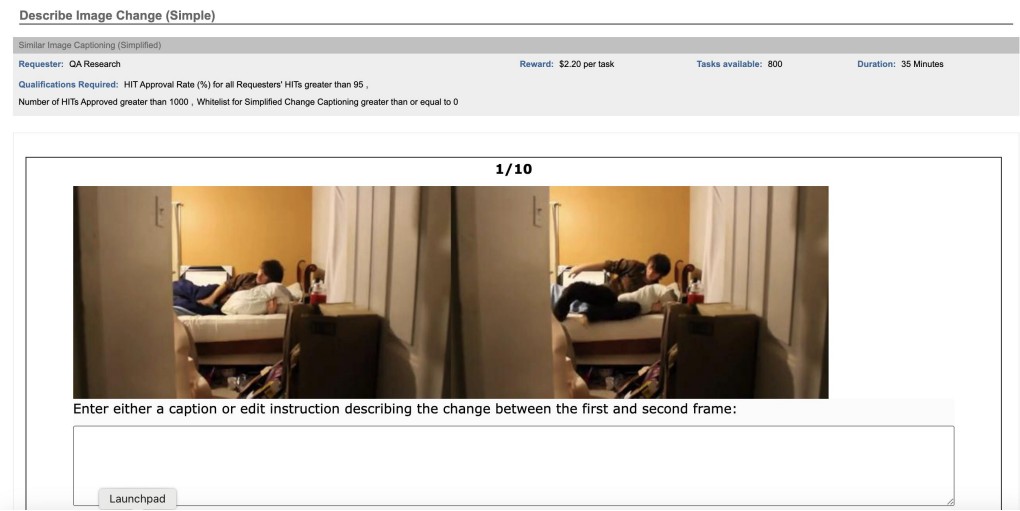

Since most of our detailed instructions and back-and-forth feedback happened via e-mail, we also provide screenshots from our e-mails:

The task:

You are still describing the difference between two images. However, two things are different now.

We want to keep it a lot simpler: So we **discard all examples where the camera moves or where more than one change happens at once.** We also discard everything that is dark/blurry. Specifically, you go through this thought process:

1. Is the camera moving at all? Write "DISCARD" (see tips below on how to quickly see whether the camera moved)
2. Is the image very blurry, very dark etc? Write "DISCARD"
3. Is there only 1 or maximally two changes happening and can you **describe them in simple short language?** If yes, write a short edit instruction! But if it is hard to describe in language and takes a lot of words because there are many changes or unusual changes, write "DISCARD" instead.

I will now give you 10 examples and show you how I want you to annotate it:

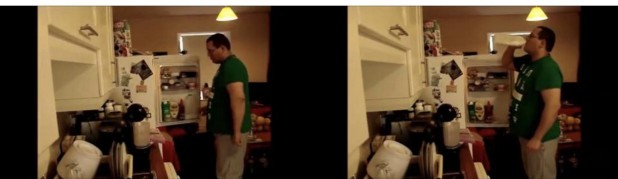

1. Is the camera moving at all? We care about even small camera movements. My tip: The best way to figure this out is by looking at the four corners and edges. In this case we can look at the bottom left corner: There is no camera movement :) That's good. So we can move to the next decision.
2. Is the image dark or blurry? Not really, so that's good too.
3. Can we describe the image in simple clear language with a single change happening? Yes :) So that means we **do not discard this example** and describe the change: **"The man lifts up the milk container and drinks it"**

## D.2 Evaluation of model outputs

For the human judgement comparing two model outputs, we mainly released our instructions as a video. Our inter-annotator agreement was a Fleiss-Kappa score of 0.626. We asked people to pay attention whether the semantics of the prompt was followed and not so much aesthetics:

Hello!

It is time to get things started.
Here are the steps:

1. https://drive.google.com/file/d/1TZu8wRJdo2lgwGdnEvxO0UyEsK0EKyJl/view?usp=sharing
**Carefully watch this video, pause sometimes to test your intuition if you would judge it like I do, before continuing to hear me explain it.**
These are the main instructions you will get!

2. **Respond after you have watched it**, and ask questions if there are some at this point.

3. Once people have responded, I will release the first 50 as a test run. **For the first 5 examples (not those I showed in the video), please in addition to annotating also write your reasoning here in the email for me to make sure you understood the task.**

4. Then I will release the rest (1K for each of you).

After this we will see if there is more to do in a week!

Thanks and best wishes,

Our AMT interface looked as follows:

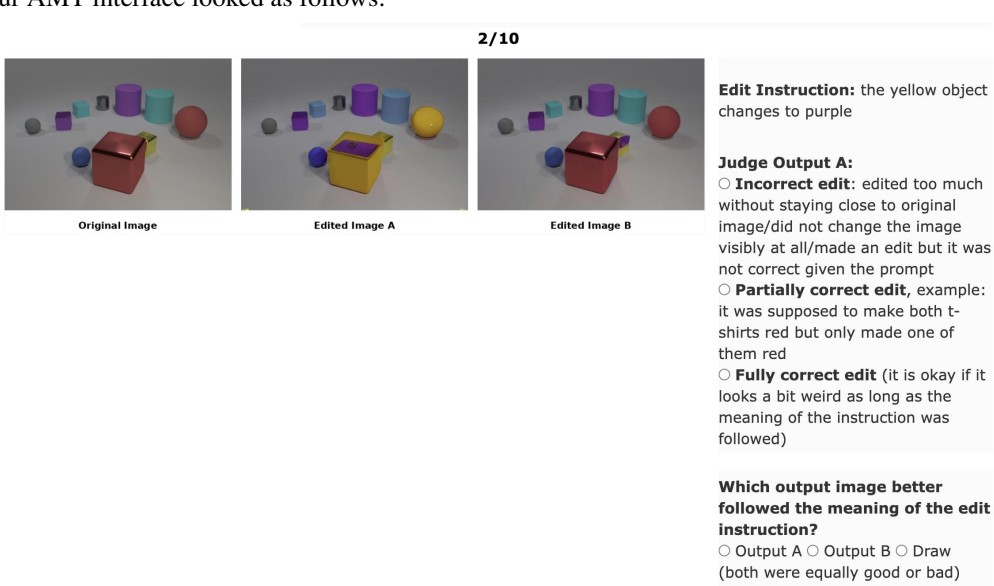

## E  Data Curation Details

**Something-Something-Edit**: We select the first and last frame of most videos from the Something Something v2 dataset Goyal et al. [2017] (most, since for some the ffmpeg software somehow couldn't retrieve the last or nth-last frame). Next, we manually went over the 174 templates that were used to create the original captions such as: "Dropping something onto something" or "Folding something", where *something* would be replaced by the respective objects. We then find certain keywords that rarely lead to useful changes and filter them out. Specifically:

{'pretend', 'holding', 'fail', 'roll', 'then letting it',
'until it falls down', 'spin', 'nothing happens', 'show',
'falling like', 'squeezing', 'throwing', 'slightly'}

**Action-Genome-Edit:** For this, we first take the original frame pairs from Wang et al. [2023] and primarily filter them for similarity, which we found to be a good proxy of camera movement. After some tuning, we settle on a cosine similarity between 0.1 and 0.4 of the two CLIP visual embeddings, allowing us to provide crowdworkers with 20K or more images; we ultimately only use 15K for this. Next, we ask humans to describe these changes and discard anything with slight camera changes,

see App. D.1 for more details. After the human filtering phase, Action-Genome-Edit contains 11K examples.

**Kubric:** These are the procedures used to generate new data for each change type: (1) *Location* changes contain one or two objects moved. They can be relative ("Move $O_1$ to the left of $O_2$"), absolute ("Move $O$ further left"), attract/repel ("Move $O_1$ and $O_2$ closer together/further apart") or swapping ("Swap the positions of $O_1$ and $O_2$") movements. (2) *Rotation* changes rotate an object around the X, Y or Z axis at 90°or 180°. Specifically, we define four rotations: turn $O$ around (Z, 180°), turn $O$ 90 degrees (Z, 90°), make $O$ fall over (X/Y, 90°), and flip $O$ upside down (X/Y, 180°). Some objects are either invariant to certain rotations (i.e. a bowl when turning it around: 180°, Z axis) or the edit is physically implausible (i.e. only vertically long objects such as cups make sense to "fall down": 90°X/Y axis). So we categorized the 213 objects into "can fall", "round" and "fully invariant" before data synthesis. (3) *Count* changes require adding or removing $n$ instances of some object. (4) *Attributes* changes vary the size, shape or color of some object. Examples for each edit in App. J.

# F   Training Details

We follow the common InstructPix2Pix setup regarding architecture and training, and rely on their code implementation [Brooks et al., 2023]. Specifically, we finetune with a batchsize of 32 and a learning rate of 5.0e-05. We also experimented with a higher resolution of 512 but saw no clear benefits (however further experiments could yield different results). We turn off the flipping augmentation for datapoints containing the word *left* or *right*, which was not an issue in previous edit training setups but would be a major issue with our focus on spatial reasoning.

Most time was spent on finding the right ratio for mixing the dataset: Our final first trains on MagicBrush alone for 13K steps and then on the full mix for 42K steps. We upsample the datasets such that they are all sampled roughly the same amount, i.e. MagicBrush and Action-Genome-Edit are multiplied by a factor of 15 while Something-Something-Edit and Kubric remain with a factor of 1. Upsampling action and reasoning-centric edits too much would result in model generations with sometimes too many or random changes. While it might've led to slightly higher numbers on those tasks, the performance on the tradtional edit tasks degraded significantly.

We had access to a total of eight NVIDIA RTX A6000 and trained models on two of them, parallelizing 4 training runs occasionally. Our training run used for the final model would run for 16 hours. We did not keep track of total compute we spent but there were many attempts at combining the datasets in different ways or training on dataset that were ultimately discarded.

# G   Extended Tables

Main tables (Tab. 2 and Tab. 5) but with additional confidence statistics (i.e. standard errors).

Table 4: Extended Table of **Human Judgment** of semantic editing success on **AURORA**-BENCH tasks. Humans were asked to rate the edit success from none (0), partial (50) to full (100). We show standard error (SE). Overall score is "balanced": we average each skill first, and then take the average of those 4 numbers.

| Model | Obj./Attr. Magic Brush | Action, Human-Object-Interaction Action-Genome | Something Something | Epic Kitchen | Reasoning WhatsUp | Kubric | CLEVR | Global Emu-Global | Overall Score |
|---|---|---|---|---|---|---|---|---|---|
| GenHowTo | $18.0 \pm 4.7$ | $8.0 \pm 2.6$ | $8.7 \pm 2.9$ | $17.7 \pm 4.6$ | $4.3 \pm 1.7$ | $0.7 \pm 0.5$ | $2.0 \pm 1.4$ | $11.3 \pm 3.1$ | $10.8 \pm 1.2$ |
| MGIE | $36.0 \pm 7.2$ | $7.0 \pm 2.4$ | $11.3 \pm 3.7$ | $5.0 \pm 1.9$ | $6.0 \pm 1.8$ | $6.7 \pm 2.8$ | $16.0 \pm 3.9$ | $36.5 \pm 6.5$ | $22.5 \pm 1.8$ |
| InstructPix2Pix | $31.3 \pm 6.8$ | $13.3 \pm 3.5$ | $12.3 \pm 4.2$ | $4.3 \pm 2.1$ | $0.7 \pm 0.7$ | $5.7 \pm 2.1$ | $14.7 \pm 3.6$ | $33.7 \pm 6.8$ | $20.5 \pm 1.8$ |
| MagicBrush | $61.7 \pm 9.7$ | $16.3 \pm 4.4$ | $17.0 \pm 4.7$ | $12.0 \pm 3.5$ | $3.0 \pm 1.7$ | $9.3 \pm 2.9$ | $22.0 \pm 4.5$ | $42.3 \pm 7.7$ | $32.6 \pm 2.4$ |
| Ours | $60.5 \pm 9.6$ | $35.6 \pm 6.6$ | $31.8 \pm 6.5$ | $14.3 \pm 4.0$ | $27.3 \pm 5.5$ | $59.6 \pm 9.3$ | $46.1 \pm 8.1$ | $33.0 \pm 6.5$ | $41.3 \pm 2.7$ |

Table 5: Extended table for DiscEdit performance with standard error (SE).

| Model | WhatsUp | Something | AG | Kubric | CLEVR |
|---|---|---|---|---|---|
| MagicBrush | $0.472 \pm 0.012$ | $0.371 \pm 0.043$ | $0.477 \pm 0.043$ | $0.392 \pm 0.045$ | $0.400 \pm 0.045$ |
| AURORA | $0.565 \pm 0.009$ | $0.548 \pm 0.045$ | $0.583 \pm 0.043$ | $0.592 \pm 0.045$ | $0.450 \pm 0.045$ |

# H   Details of *DiscEdit*

Our *DiscEdit* metric takes an images and two similar prompts $t_{nochange}$ and $t_{change}$, where the model is expected to not change anything given $t_{nochange}$ but change something (i.e. normal edit) with $t_{change}$. Practically, one can take an existing image-edit pair dataset and either automatically or manually come up with "no-change" prompts.

Here we illustrate this process for some of **AURORA**-BENCH examples. Note that some edits do not have a straightforward associated "no-change", most notably adding objects which are many MagicBrush edits. For example, "add a squirrel next to the lamp" does not have any equivalent "no-change". Let's imagine there is already a dog next to the lamp; so could we have "add a dog next to the lamp" as $t_{nochange}$? No, since it would be a correct model response to then add a second dog. So here are examples for several **AURORA**-BENCH tasks, with the goal of illustrating the point of changing *more or less relatively speaking*, not to show cherry-picked perfect examples - WhatsUp, Something Something, AG, Kubric, CLEVR:

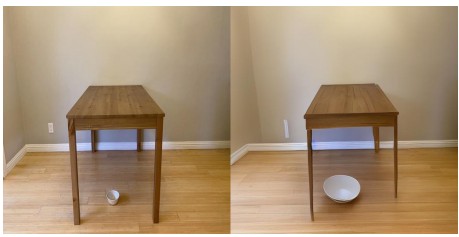 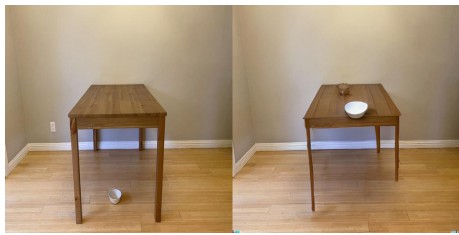

(a) No-change prompt: *Move the bowl under the table*    (b) Change prompt: *Move the bowl on the table*

Figure 11: WhatsUp

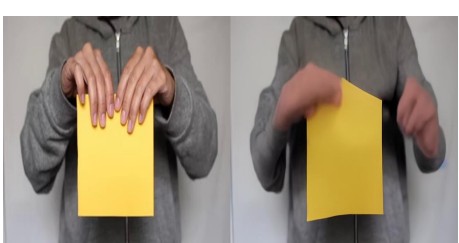 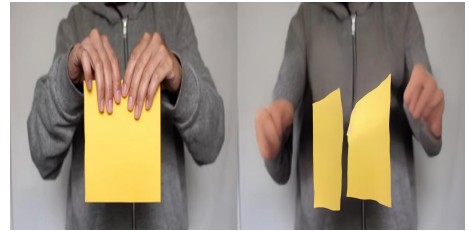

(a) No-change prompt: *Keep the paper intact*    (b) Change prompt: *Tear the paper into two pieces*

Figure 12: Something Something

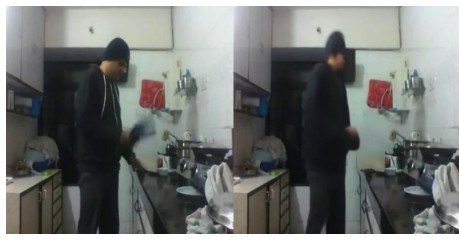 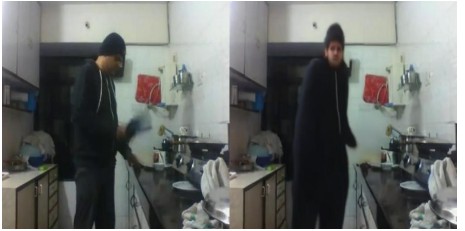

(a) No-change prompt: *Make him face to the right*    (b) Change prompt: *Make him face towards the camera*

Figure 13: Action Genome

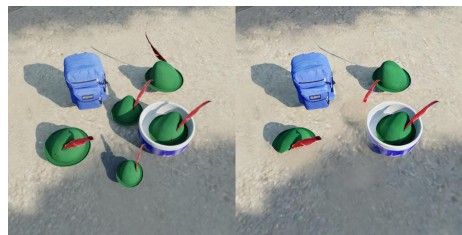 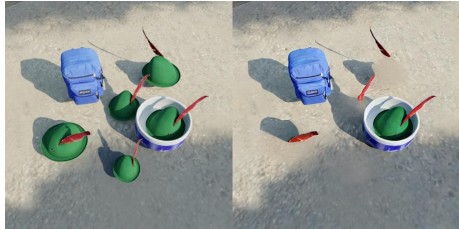

(a) No-change prompt: *Remove all yellow objects*

(b) Change prompt: *Remove all green objects*

Figure 14: Kubric

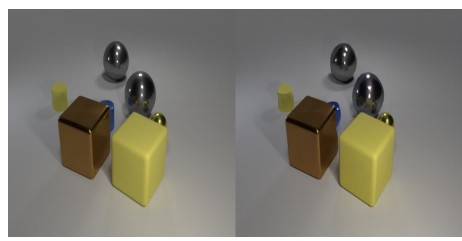 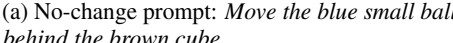 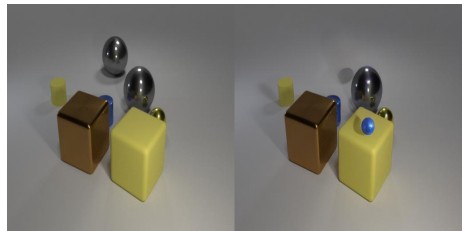

(a) No-change prompt: *Move the blue small ball behind the brown cube*

(b) Change prompt: *Move the blue small ball in front of the brown cube*

Figure 15: CLEVR

## H.1 Implementation details

For each triplet of (image, change-prompt, no-change-prompt), we start with the same initial noisy latent for both minimally different prompts to reduce variance, a common procedure in using text-to-image models as discriminators [Krojer et al., 2023, Li et al., 2023]. To have a larger sample size, and thus reduce variance further, we also sample four different noisy latents per triplet and treat it is as separate examples when computing the accuracy.

We end up using 30 examples for Something-Something-Edit, Action-Genome-Edit, Kubric and CLEVR, and all 800 examples from WhatsUp. WhatsUp was the only task that needed no manual writing or filtering of minimally different prompts due to its well-defined spatial movement setup.

## I Sample generations from our evaluation

### I.1 Samples used for human judgement

We show four randomly sampled outputs from our **AURORA** model on each of the eight **AURORA-BENCH** tasks.

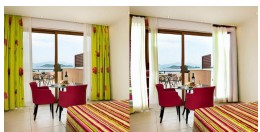 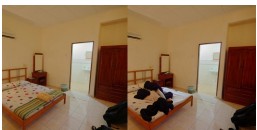 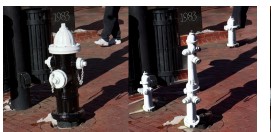 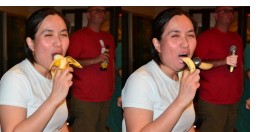

(a) Prompt: *change the bright lime green curtains to white curtains*

(b) Prompt: *Let the blanket be longer and hang over the bed.*

(c) Prompt: *Make the hydrant all white*

(d) Prompt: *Change the banana to a microphone.*

Figure 16: **MagicBrush**

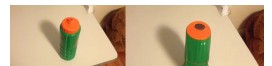 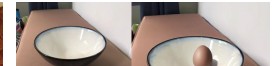 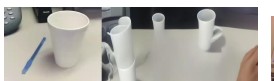 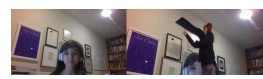

(a) Prompt: *Flip the bottle upside down*    (b) Prompt: *Putting egg into the bowl*    (c) Prompt: *Moving cup away from pen*    (d) Prompt: *Lift her hands up to show a piece of paper*

Figure 17: **Something-Something-Edit**

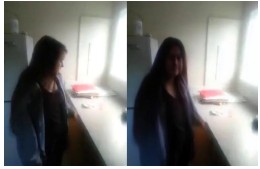 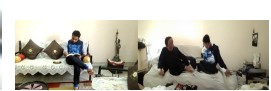 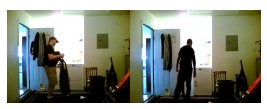 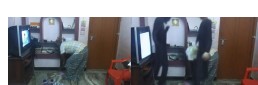

(a) Prompt: *Make her face fully visible looking at the camera*    (b) Prompt: *Make the man put down the book on the sofa*    (c) Prompt: *Make him walk further towards the right*    (d) Prompt: *Make them stand up*

Figure 18: **Action-Genome-Edit**

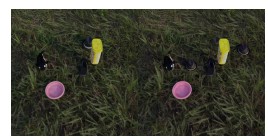 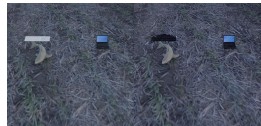 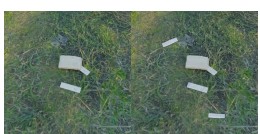 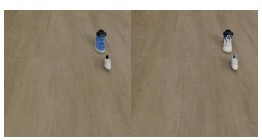

(a) Prompt: *place the black hat on the right hand of the yellow nesquik chocolate powder canister*    (b) Prompt: *convert the white keyboard into a black keyboard*    (c) Prompt: *add 1 white keyboard to the platform*    (d) Prompt: *turn the blue shoe into a white running shoe*

Figure 19: **Kubric-Edit**

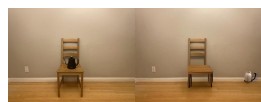 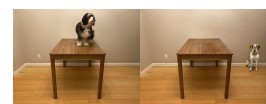 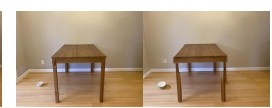 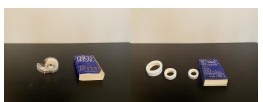

(a) Prompt: *Move the kettle to the right of the chair*    (b) Prompt: *Move the dog to the right of the table*    (c) Prompt: *Move the bowl under the table*    (d) Prompt: *Move the book behind the tape*

Figure 20: **WhatsUp**

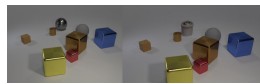 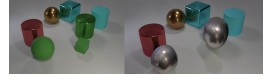 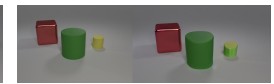 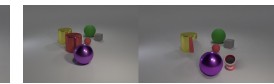

(a) Prompt: *the cylinder becomes gray*    (b) Prompt: *the big metal sphere becomes gray*    (c) Prompt: *make the big green rubber cylinder turn purple*    (d) Prompt: *move the red cylinder to the right of the gray block*

Figure 21: **CLEVR**

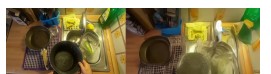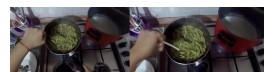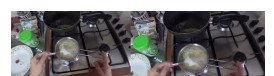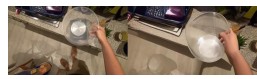

(a) Prompt: *Grab the soap bottle on the right with the hand*
(b) Prompt: *Pull one noodle out of the pot with the fork*
(c) Prompt: *Move the right hand to one of the oven knobs*
(d) Prompt: *Drop the transparent large bowl onto the floor*

Figure 22: **EpicKitchen**

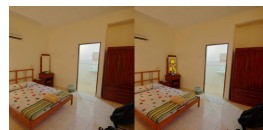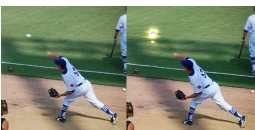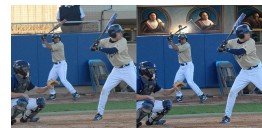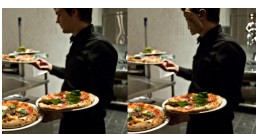

(a) Prompt: *Make this image look like the Simpsons Cartoon.*
(b) Prompt: *Change the image to be taking at night.*
(c) Prompt: *Make it look like a renaissance painting.*
(d) Prompt: *Give this image a look inspired by Michelangelo's "David".*

Figure 23: **Emu**

## I.2 For DiscEdit

See App. H

## J  Random (=non-cherry picked) Samples from existing and contributed training datasets

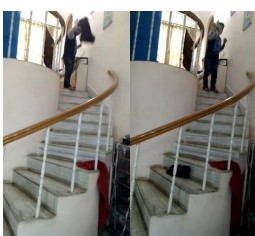

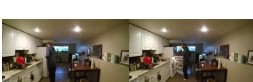

Make the man open the refrigerator

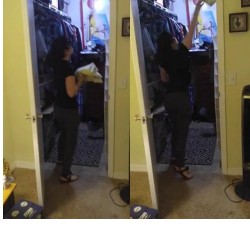

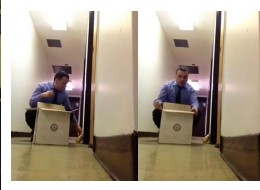

The man has turned his face to the left to look into the box.

Throw the black clothes down to the stairs

Lift the hand to place the clothes on the top

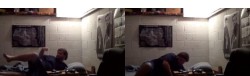

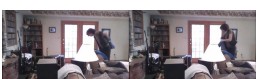

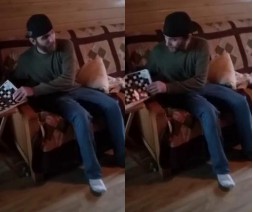

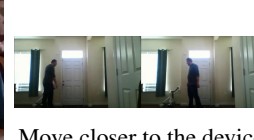

Move closer to the device from the right side

Hold the ber and take legs down to rise

Put on the jacket

Turn the face to the left on the book

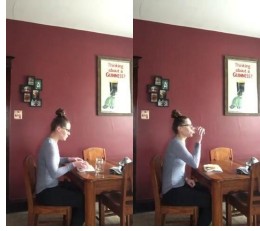

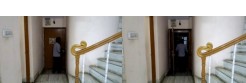

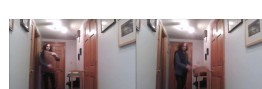

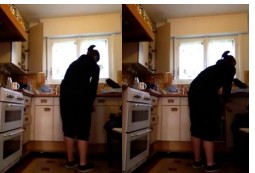

Open the door and move further

The man placed the pillow on the table next to him.

Bend a bit towards the desk

Put the left hand on the body while drinking with the glass

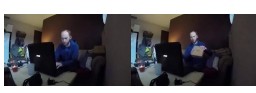

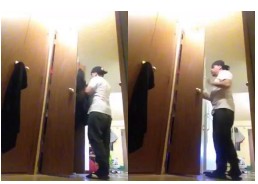

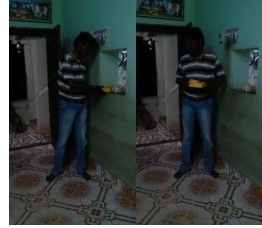

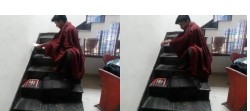

The person has dropped a white object on a stair above him.

The man has picked up a brown object in his hands.

Keep the jacket in the wardrobe

The man has picked up a yellow object from the shelf.

**Figure 24:** 16 randomly sampled training datapoints from Action-Genome-Edit

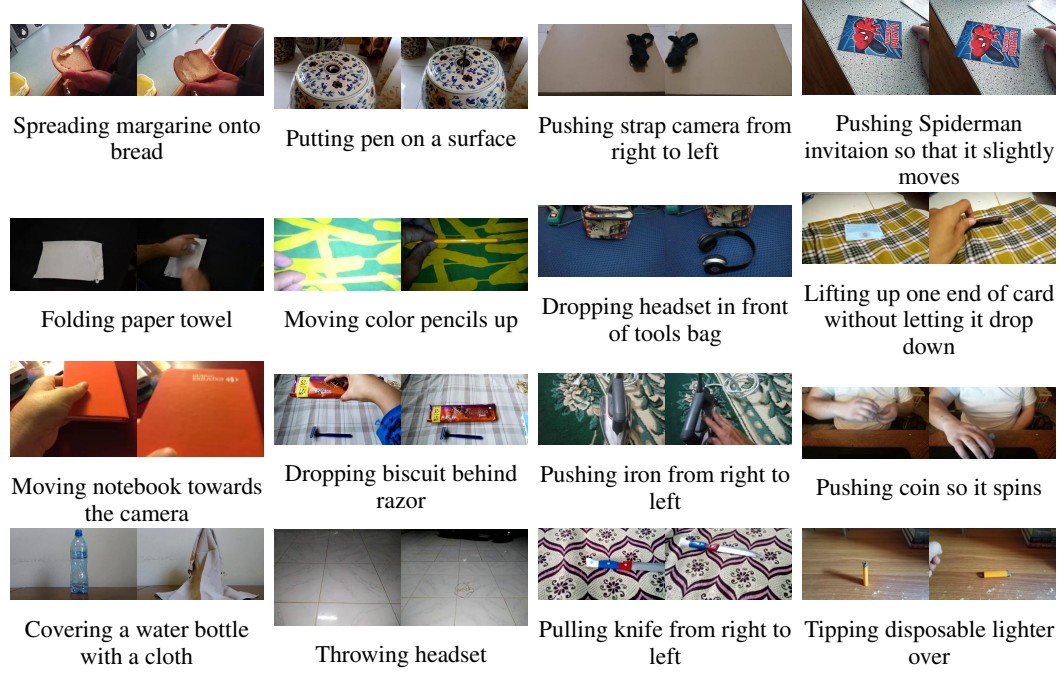

**Figure 25:** 16 randomly sampled training datapoints from Something-Something-Edit.

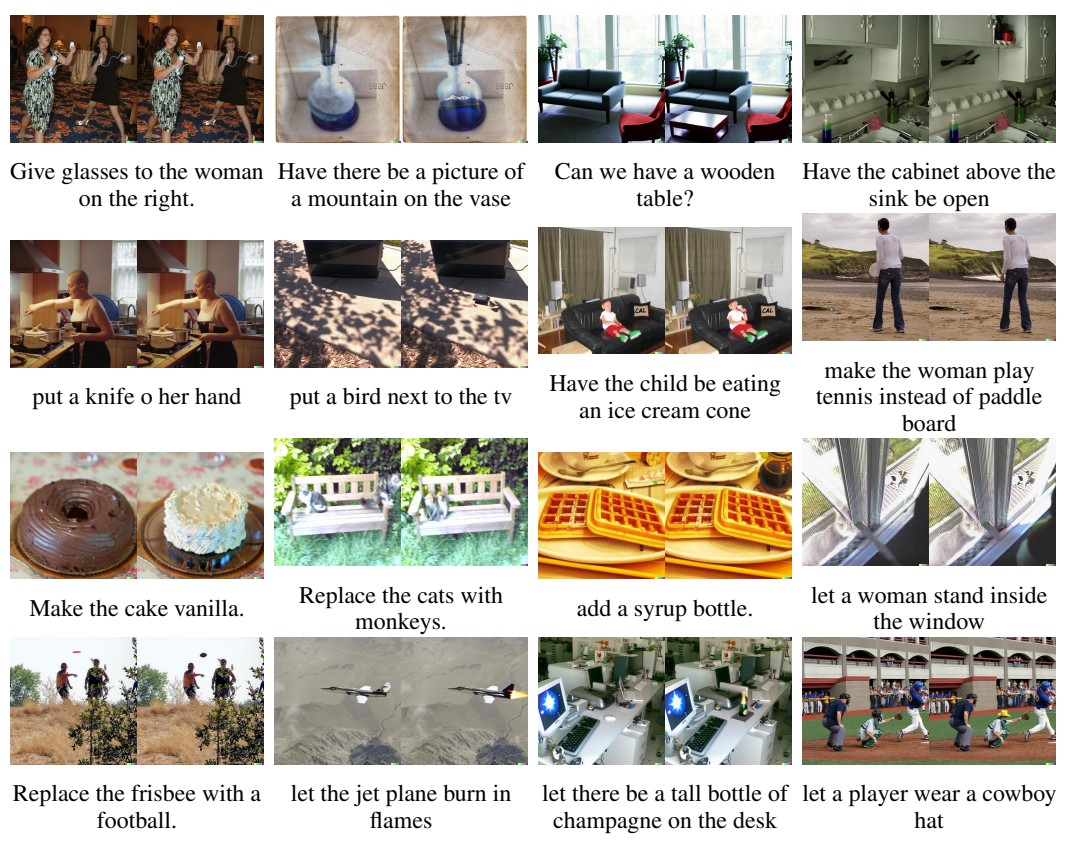

**Figure 27:** 16 randomly sampled training datapoints from MagicBrush.

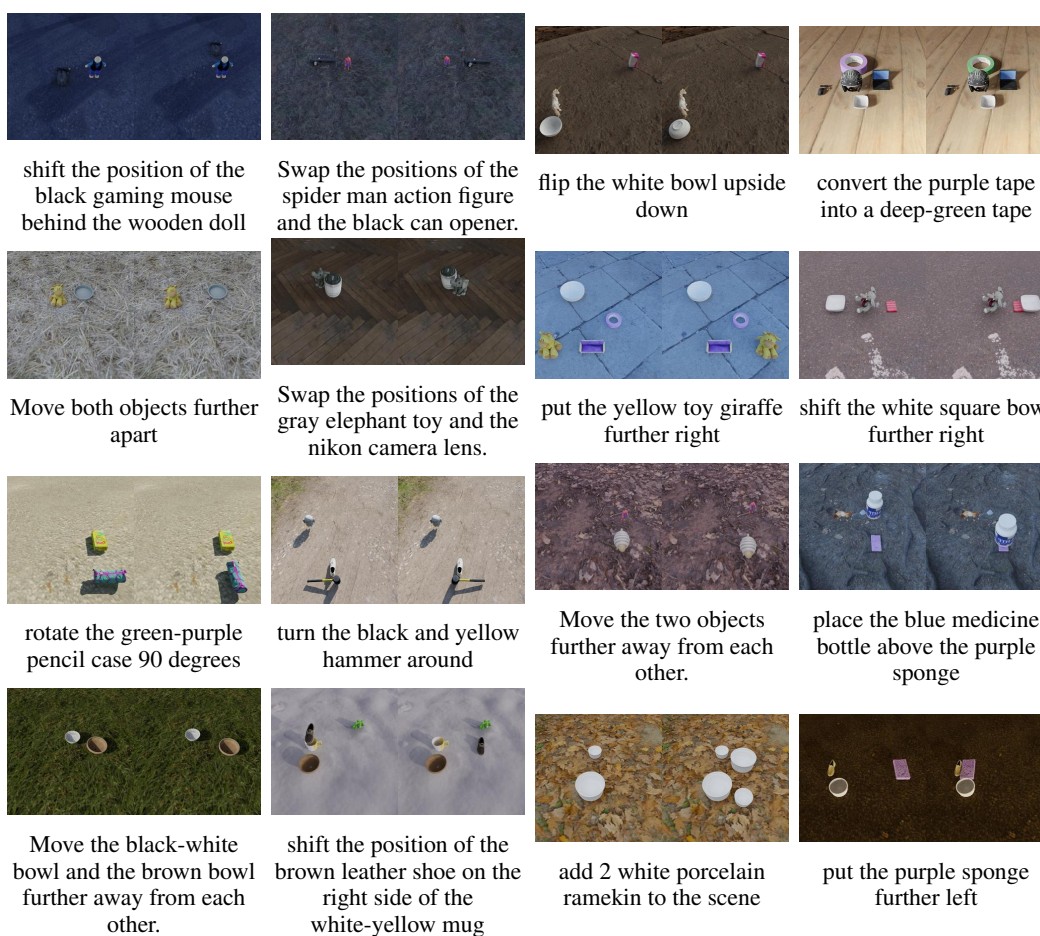

**Figure 26:** 16 randomly sampled training datapoints from Kubric-Edit dataset.

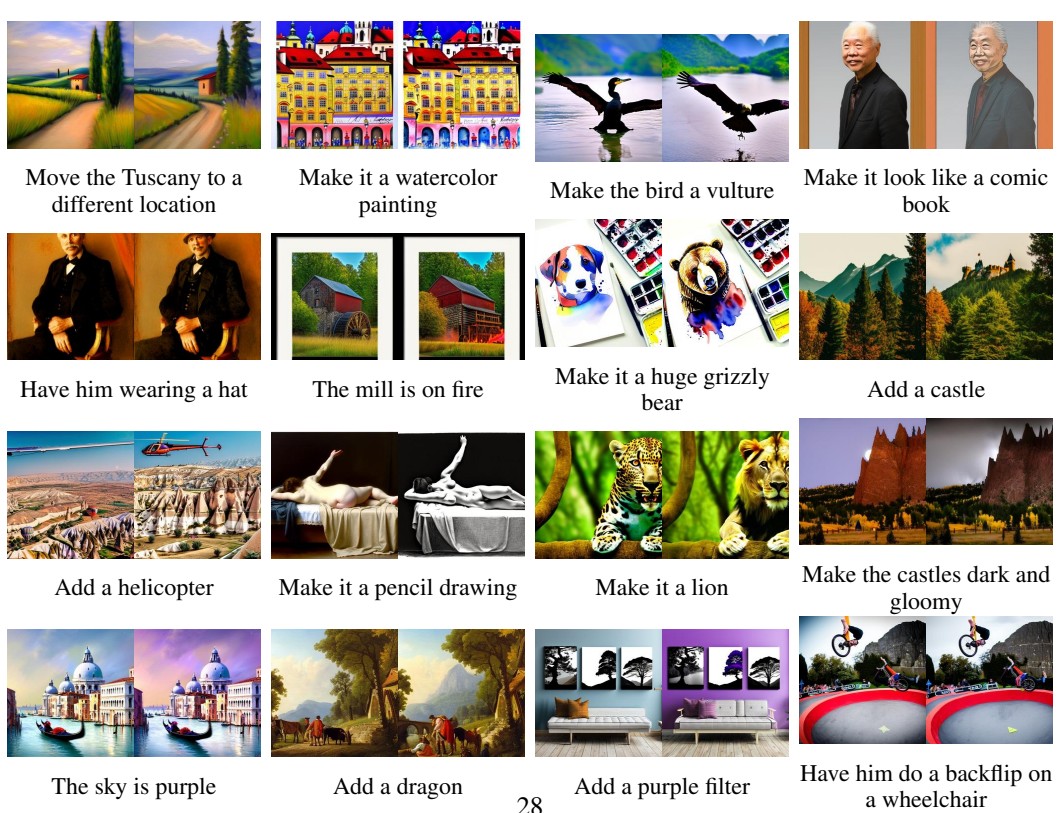

**Figure 28:** 16 randomly sampled training datapoints from InstructPix2Pix

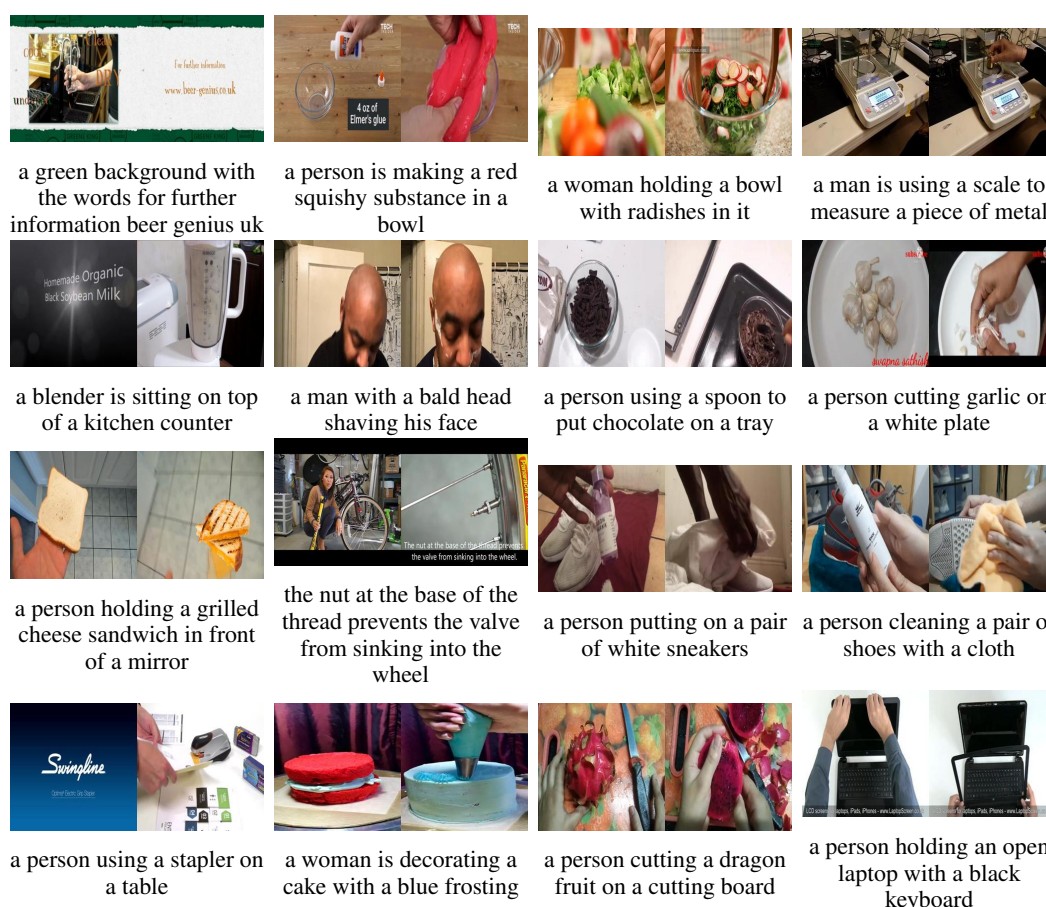

**Figure 29:** 16 randomly sampled training datapoints from GenHowTo dataset.

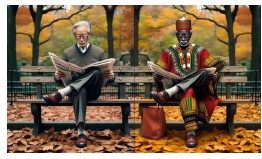

change the person's attire to include a colorful dashiki garment with intricate patterns, a matching kufi cap, and add a brown leather bag beside them on the bench

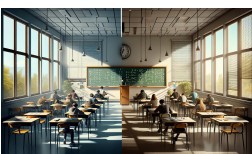

add students taking an exam with papers and pencils on their desks

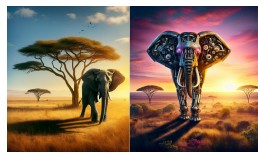

Replace the elephant's organic body with mechanical parts, giving it the appearance of a robot while maintaining its original pose and position in the savanna.

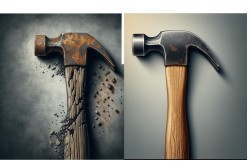

restore the hammer by removing rust, polishing the metal, and replacing the handle with a new wooden one

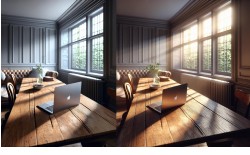

Increase the intensity of sunlight coming through the windows to brighten the room significantly.

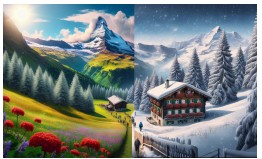

transform the scene to a snowy winter landscape

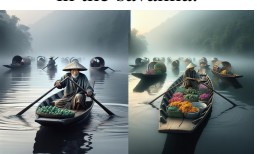

add various colorful flowers, fruits, and vegetables to the boat, and include additional boats with similar items in the background to create a floating market scene

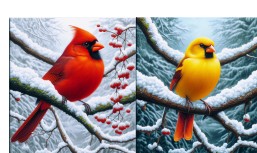

The color of the cardinal bird has been changed from red to yellow, and the tail feathers have been altered from red to a gradient of yellow to red.

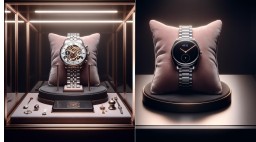

Replace the mechanical watch with a smartwatch, remove all accessories, and simplify the display to only include the smartwatch on the plush pillow stand with a dark background.

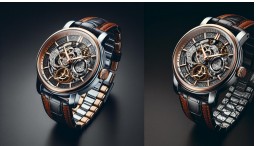

EDIT OPERATION DESCRIPTION: The watch's metal bracelet has been edited to include a cracked texture design, giving the appearance of a damaged or stylized surface.

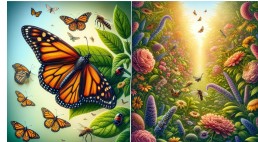

Replace the simple leafy background with a vibrant and colorful garden scene, filled with a variety of flowers like roses, daisies, and lupines, and add more butterflies and bees to create a lively atmosphere.

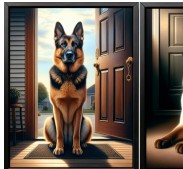

Replace the German Shepherd dog with a Siamese cat.

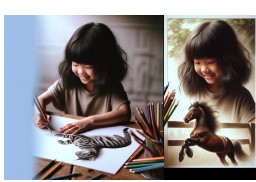

Replace the drawing of the tabby cat with a drawing of a galloping brown horse.

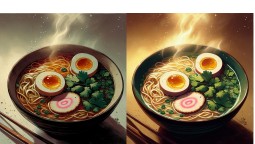

Change the color of the bowl's rim from dark brown to green.

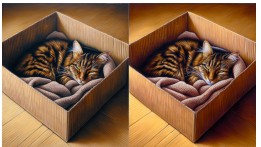

Remove the cat's eyes and make them closed as if it is sleeping.

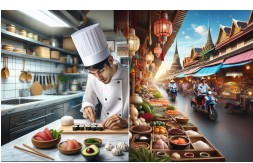

Replace the well-equipped kitchen background with a vibrant street food market scene, complete with stalls, hanging lanterns, and a bustling crowd.

**Figure 30:** 16 randomly sampled training datapoints from HQ-Edit dataset.

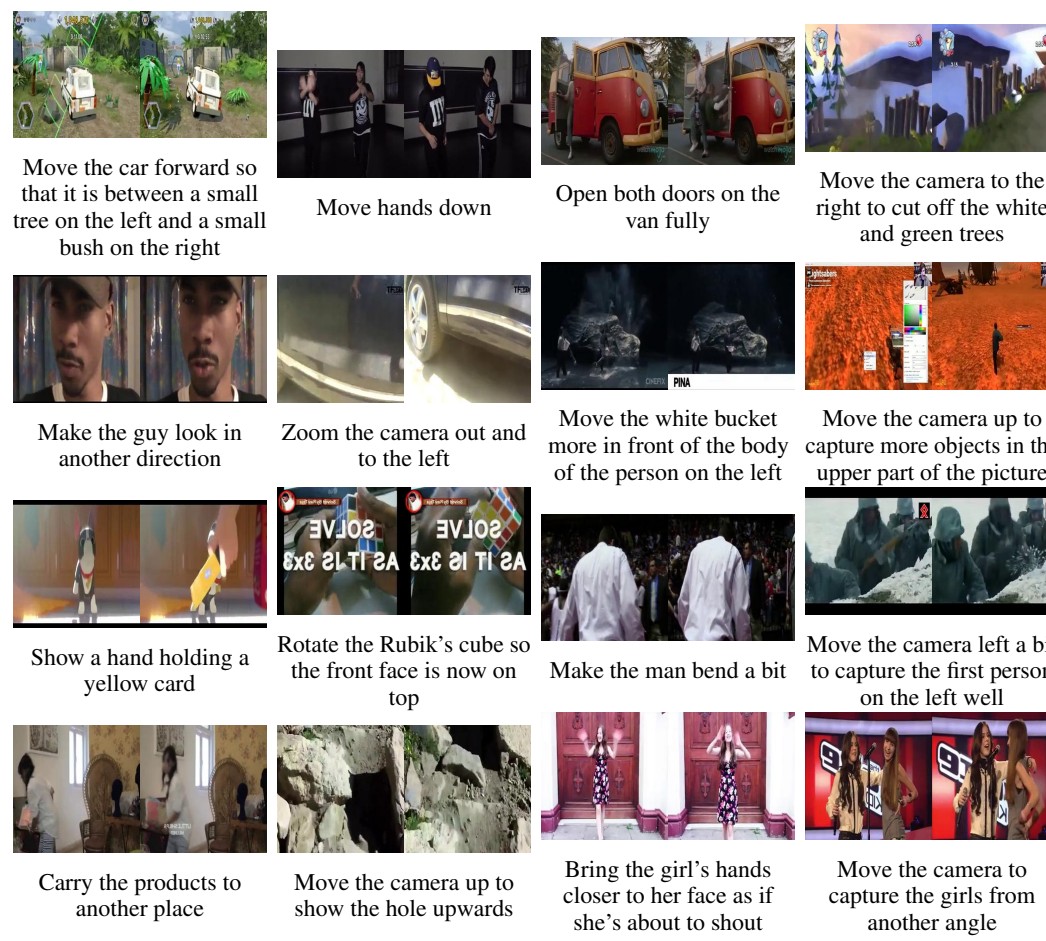

**Figure 31:** From a dataset we collected similar to Action-Genome-Edit but discarded in the end: 16 randomly sampled training datapoints from MSR-VTT-Edit dataset.

# K    Details of EditDAAM

In this section, we try to go in more detail into Sec. 5.4 (EditDAAM). The main goal is to answer how the model understands the input image and instruction and how the input image is manipulated to generate the final image based on the instruction. For this we analyze the attention maps the model and found out that the majority of the understanding and reasoning occurs during the first iteration of the diffusion process. It is at this stage that the model determines the appropriate course of action. In subsequent iterations, the model focuses on localization and refine upon the reasoning established in the previous steps.

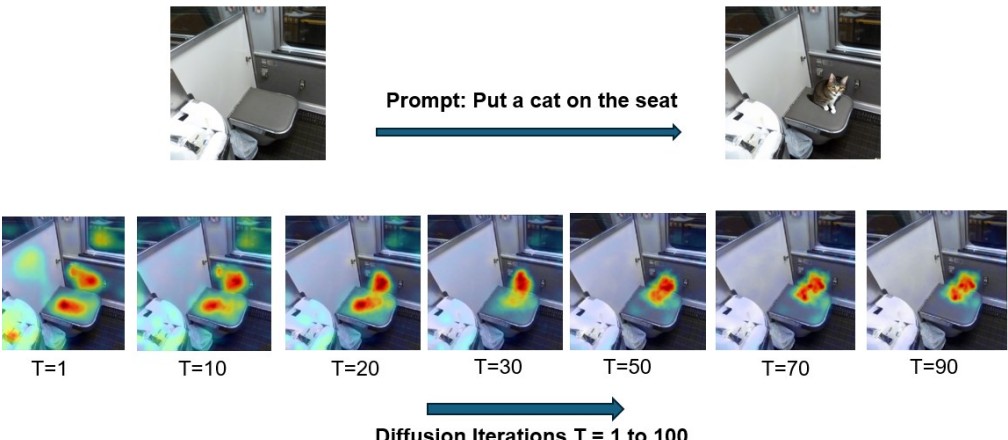

T=1    T=10    T=20    T=30    T=50    T=70    T=90

**Diffusion Iterations T = 1 to 100**

Figure 32: For the prompt 'Put a cat on the seat,' we observe that crucial decisions are made at the initial timestep (T=1), and subsequent iterations progressively refine the heatmaps, enhancing the model's understanding based on previous iteration.

To understand the crucial first iteration, we analyze the U-Net architecture's three modules: down, middle, and upper. Our findings show that the down and middle modules are responsible for comprehending the input image and localizing objects based on the prompt. The upper module then performs the reasoning task, deciding where to make changes or place objects according to the instruction. Below represents the images Below represents the figure which shows the Input image and its corresponding generated image. Down layers represent the average of all attention maps present in the down module in the first iteration of the diffusion process. The middle layers and upper layers represent the average of attention maps in their respective modules in the first iteration.

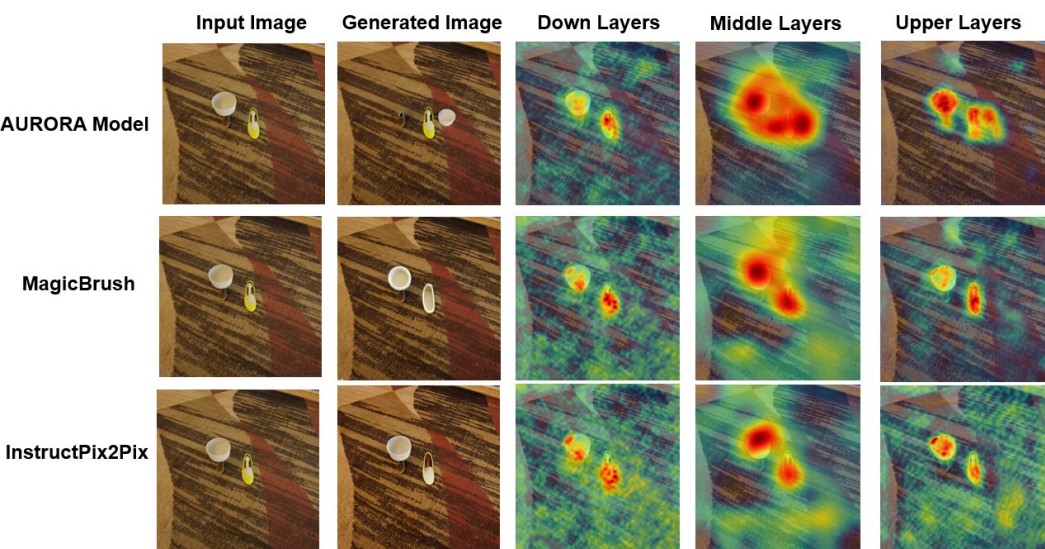

Figure 33: **Prompt:** *Put the white porcelain ramekin on the right hand of the brown shoe.* We show attention maps for three levels of U-Net layers: Down, Middle, Upper.

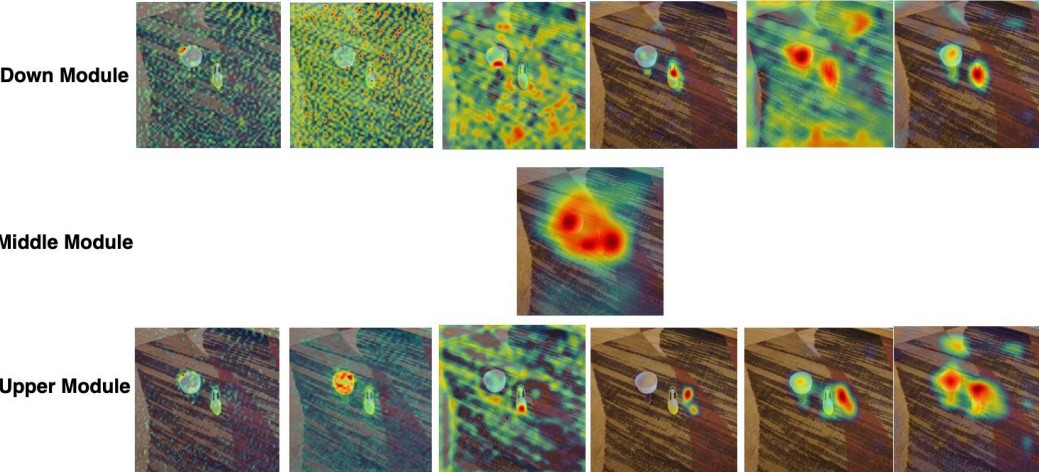

Figure 34: **Prompt:** put the white porcelain ramekin on the right hand of the brown shoe. **Down Module:** The model interprets salient low-level features and gradually recognizes image parts based on the given prompt. **Middle Module:** This module localizes all objects mentioned in the prompt, which are necessary for reasoning. **Upper Module:** This module aggregates information from the down and middle modules and performs spatial reasoning. In the forth upper module heatmap, the model concentrates on the right side of the shoe, where it intends to place the object, consistent with the prompt. We observe that Zhang et al. [2024] and Brooks et al. [2023] lack this kind of spatial reasoning.

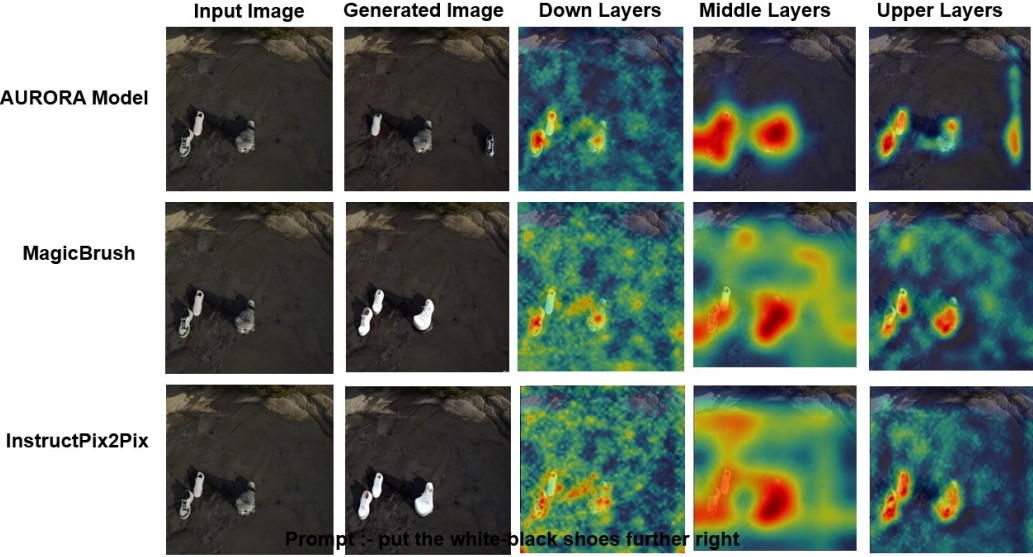

Figure 35: **Prompt:** put the white-black shoes further right. **Input Image:** The image consists of shoes and a cat. The prompt is to move the white-black shoe to the right.**Generated Image:** We can observe that the **AURORA** model accurately comprehended the prompt and positioned the white-black shoe correctly, outperforming Zhang et al. [2024] and Brooks et al. [2023].**Down Layers:** This heatmap represents the average attention maps of the down module, indicating that all three modules attempt to understand the input image.**Middle Layers:** It represents the heatmap of the middle modules, which localize all the objects present in the prompt. **Upper Layers:** This represents the average of the upper module, and we can see that the **AURORA** model demonstrates a good spatial understanding and decides where to place the object, as evident in the heatmap, unlike Zhang et al. [2024] and Brooks et al. [2023], which fail to comprehend the prompt accurately.

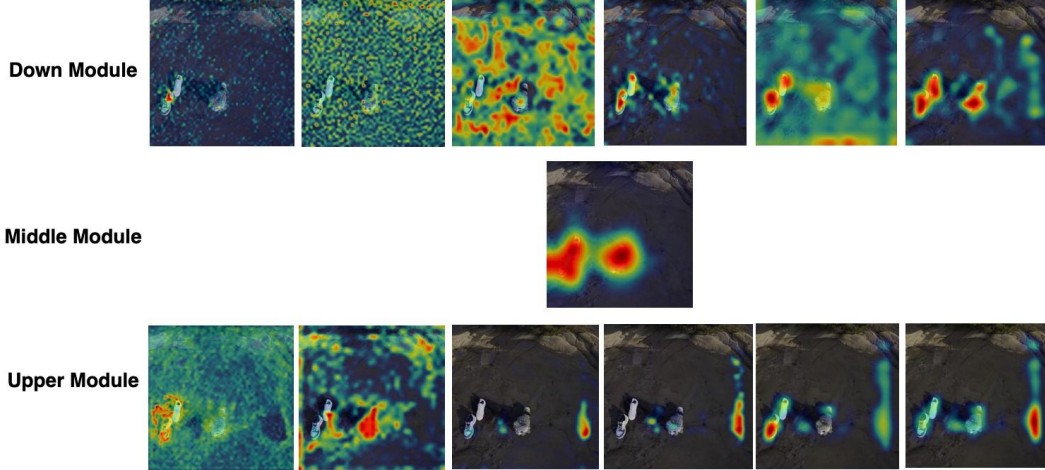

Figure 36: **Prompt:** Put the white-black shoes further to the right. **Down Module:** The model interprets salient low-level features and gradually recognizes image parts based on the given prompt. **Middle Module:** This module localizes all objects mentioned in the prompt, which are necessary for reasoning. **Upper Module:** This module aggregates information from the down and middle modules and performs spatial reasoning. In the third upper module heatmap, the model concentrates on the right side, where it intends to place the object, consistent with the prompt. We observe that Zhang et al. [2024] and Brooks et al. [2023] lack this kind of spatial reasoning.

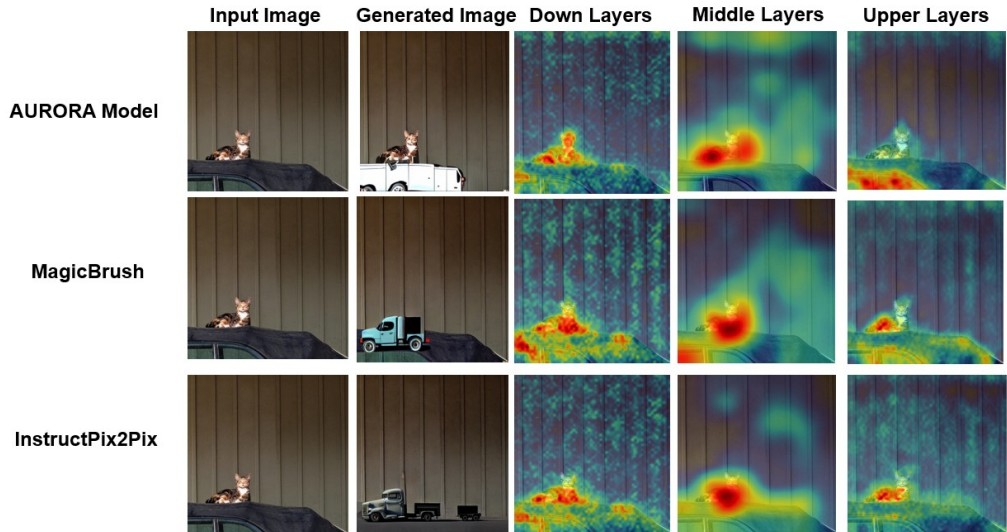

Figure 37: **Prompt:** change the car to a truck. **Input Image:** The image consists of a cat and a car. The task is to change the car to truck. **Generated Image:** Only AURORA is able to generate the right image which follows the prompt. **Down Layers:** The module attempts to understand the input image, focusing on the cat and car. **Middle Layers:** This heatmap shows the middle module localizing both objects mentioned in the prompt: the cat and the car to be changed. **Upper Layers:** The average heatmap of the upper module shows AURORA demonstrating strong reasoning, deciding where to place the truck in relation to the cat, outperforming other models that fail to accurately interpret the prompt.

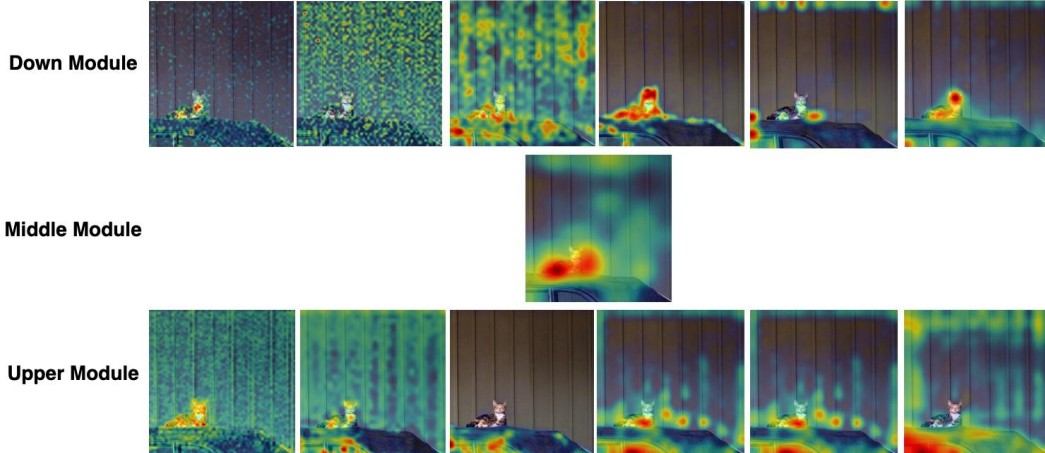

Figure 38: **Prompt:** change the car to a truck. **Down Module:** The model interprets salient low-level features and gradually recognizes image parts based on the given prompt. **Middle Module:** This module localizes all objects mentioned in the prompt, which are necessary for reasoning. **Upper Module:** This module aggregates information from the down and middle modules and performs spatial reasoning. In the third upper module heatmap, the model concentrates on the right side, where it intends to place the object, consistent with the prompt. We observe that Zhang et al. [2024] and Brooks et al. [2023] lack this kind of spatial reasoning.

## L   Comparison of most common verbs and nouns

The following analysis (see Fig. 39) was conducted for the rebuttal and here is our explanation from the rebuttal itself:

*"The remaining limitation is an understanding of the landscape of actions and reasoning that can be considered. For example, there are many many verbs that one could use in a reasoning context – even visual genome has a large number of verbs"*: It would be fascinating to see if there is a different distribution of verbs between broad generic VL captioning datasets and datasets tailored for action-editing! Even before conducting further experiments, we can already say that a lot of edits have the generic verbs like "move", which is nonetheless often still a complex task: "Move the cup closer to the plate" is a very different move than "Move the hand closer to their hair", where the exact action required is implicit in the scene/affordances/angles and not reflected in the textual "move". I think this is somewhat inherent to the editing task itself where captions tend to be shorter than traditional caption datasets, often with simpler verbs "make OBJECT ATTRIBUTE" (make the horse darker) or "replace/add OBJECT". So another interesting comparison would be the distribution of verbs in traditional (object/attribute) editing vs. our focus on action editing. Finally, we note that a lot of complexity in our data comes from other linguistic constituents such as prepositions or adjectives/adverbs, e.g. "Move the hand slightly closer under the table with the finger pointing upward" where slightly, closer, under, upward are all interesting to understand but not verbs. To study the frequency differences to established caption datasets, we visualize the verb and noun distribution in MS-COCO, AURORA as well as the four subsets of AURORA. See PDF figure caption for the details! Overall, the verbs are less diverse but as described above a lot of the complexity comes from other textual or visual aspects. On top, the verbs are quite different to COCO and notably also quite different to more established object/attribute-centric editing. Also note that while "make" is a very frequent verb, it can often be accompanied with one of the other verbs like "make the person stand up".

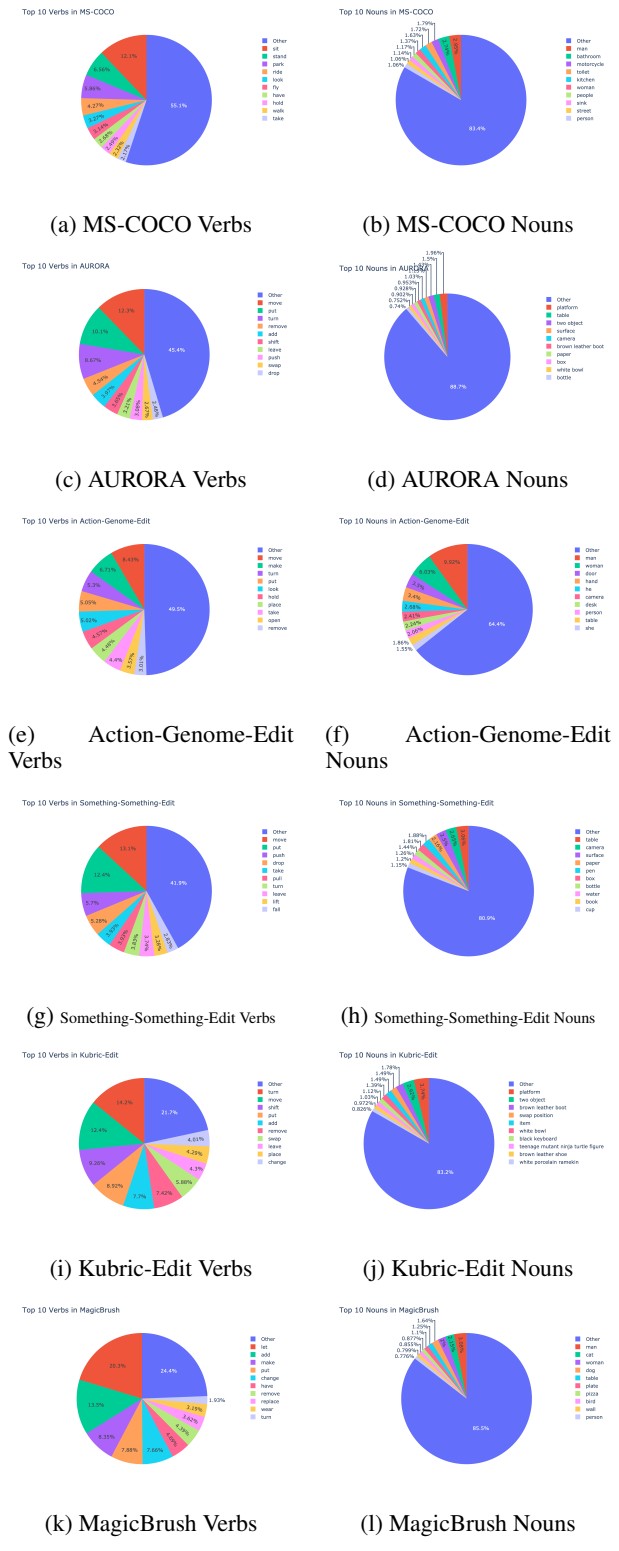

(a) MS-COCO Verbs

(b) MS-COCO Nouns

(c) AURORA Verbs

(d) AURORA Nouns

(e)    Action-Genome-Edit Verbs

(f)    Action-Genome-Edit Nouns

(g) Something-Something-Edit Verbs

(h) Something-Something-Edit Nouns

(i) Kubric-Edit Verbs

(j) Kubric-Edit Nouns

(k) MagicBrush Verbs

(l) MagicBrush Nouns

Figure 39: **Frequencies of verbs and nouns in caption and instruction-editing datasets.** We draw attention to three comparisons: First, how MS-COCO verbs (a) differ from our collection of editing datasets (c). Second, how our action-centric editing verbs (e,g,i) differ from object/attribute-centric editing (k). Third, how the nouns are more similar between caption and editing datasets overall (except for synthetic Kubric and Something-Something).

# M  Behind the scenes

In this section we want to document the whole research process and not just the final product. Specifically, we show how the paper went from first idea to what you read here. By doing this, we also show the things that did not work and might be insightful for other researchers.

We started the project in January 2024 coming out of the Christmas holidays: The first author had realized that another project wasn't going anywhere [5], and was going through some other ideas with the more senior authors. We settled on the broad direction of "How can we leverage nearby video frames for image editing¿' because it seemed like learning from videos for these sorts of tasks is the next step that not many have explored, and because it fitted into the story of previous PhD papers of the first author.

Most of January, February and March were spent scouting various datasets:

## M.1  Discarded/unsuccessful datasets

Simply repurposing change caption or contrastive datasets [Park et al., 2019, Wang et al., 2023, Krojer et al., 2022] that contain two images and some captions of how they are different is not enough: To give one example, change captions are often underspecified ("the car changed its location", "the man is not touching the shelf") with nondescript prompts or images containing more changes than described in prompts.

We also experimented with the initial pre-training stage on noisier but large scale data such Instruct-Pix2Pix, i.e. adding noisier video-data or HQ-Edit to the mix. But we found these datasets not very helpful via manual inspection of outputs: they often diverge strongly from the source image, or artifacts of automatic data curation (GenHowTo)

We came across Michel et al. [2023] too late. It might have either complemented or even replaced Kubric for the synthetic generation part.

We even collected almost 10K change/edit descriptions on top of nearby frames from MSR-VTT videos [Xu et al., 2016], see Fig. 31 for 16 samples from this collected data. At that point, in late March we finally began to realize how much harder this task was than we initially expected: Most videos are not really suitable, so even with high-quality human annotation it won't lead to a strong editing model. Videos are hard to learn from!

Initially we had set out to build a very general editing model, thus looking at very general video datasets from all of YouTube or lots of movies. We ended up using narrower videos depicting well-defined actions since videos in the wild do not depict meaningful edits. This narrowed scope was initially frustrating but ultimately led to a better paper.

## M.2  Reflections on choosing and managing a research project (from a first person perspective of the first author)

This project taught me how important it is to really really define your research question and contribution as early as possible, and for that it helps knowing the literature and what it means to contribute to science. I should know early whether my main contribution is methodological, a new training dataset or better evaluation, or something else! From there, I should have one (or maybe two) clear research question in mind (and not five!). This doesn't mean there can be other questions but ideally they should be sub-questions of the main one and emerge as ablation studies later on. Most of these lessions were painfully learned during the final writing which becomes very hard when there is too many things you wanted to contribute and the project was many different things at different points.

## M.3  Tips for anyone working on something similar

1. Learning from videos is hard so don't naively assume large-scale YouTube will solve anything. Lots of curation is needed.

---

[5]It was one of those interesting toy problem that is fun to work on but too nice to attract much attention probably.

2. I am curious if video generation is a more viable approach to the edits we are looking at. However it is more costly if all you care about is the edit so maybe there is some way to compress/distill the video model into an editing model?

3. Good metrics are almost non-existant. Please don't use old ones simply because someone else did. Rely on human ratings, and do them well. Let's build better metrics!

4. Image editing seems like a more intriguing problem for people interested in vision-and-language reasoning, compared to the more mainstream text-to-image generation.

# N Datasheet for Dataset "AURORA"

## N.1 Motivation

*The questions in this section are primarily intended to encourage dataset creators to clearly articulate their reasons for creating the dataset and to promote transparency about funding interests.*

### N.1.1 For what purpose was the dataset created?

We collected AURORA since there is no current high-quality dataset for instruction-guided image editing where the instruction is an action such as "move carrots into the sink". This is an important subtask of image editing and can enable many downstream applications.

### N.1.2 Who created the dataset (e.g., which team, research group) and on behalf of which entity (e.g., company, institution, organization)?

It was developed primarily at "Mila - Quebec Artificial Intelligence Institute", specifically in Siva Reddy's lab by his PhD student Benno Krojer.

### N.1.3 Who funded the creation of the dataset?

The dataset was funded by the PI, Siva Reddy.

### N.1.4 Any other comments?

None.

## N.2 Composition

*Most of these questions are intended to provide dataset consumers with the information they need to make informed decisions about using the dataset for specific tasks. The answers to some of these questions reveal information about compliance with the EU's General Data Protection Regulation (GDPR) or comparable regulations in other jurisdictions.*

### N.2.1 What do the instances that comprise the dataset represent (e.g., documents, photos, people, countries)?

Each datapoint is a triplet of (source image, prompt, target image), i.e., (an image of a dog, "make the dog smile", an image of a dog smiling).

### N.2.2 How many instances are there in total (of each type, if appropriate)?

There are 399K instances in total, distributed across four sub-datasets.

### N.2.3 Does the dataset contain all possible instances or is it a sample (not necessarily random) of instances from a larger set?

It is not really a sample, but we did have to filter out video frames that were too noisy or showed too much change such as camera movement.

### N.2.4 What data does each instance consist of?

Two raw images (source & target), and a string (the prompt).

### N.2.5 Is there a label or target associated with each instance?

The target image is the structure that the model has to predict during training and test time.

### N.2.6 Is any information missing from individual instances?

No.

### N.2.7 Are relationships between individual instances made explicit (e.g., users' movie ratings, social network links)?

In MagicBrush there are sequential edits, that are indicated by the json key "img_id".

### N.2.8 Are there recommended data splits (e.g., training, development/validation, testing)?

We release training data separately from the AURORA-Bench data. The test data is much smaller and the test split of each of our training sub-datasets contributes to it.

### N.2.9 Are there any errors, sources of noise, or redundancies in the dataset?

The main source of noise comes with the video-frame-based data where sometimes there can be more changes than described in language.

### N.2.10 Is the dataset self-contained, or does it link to or otherwise rely on external resources (e.g., websites, tweets, other datasets)?

Self-contained, except that we ask people to download the videos from the original Something Something website, instead of providing the actual image files.

### N.2.11 Does the dataset contain data that might be considered confidential (e.g., data that is protected by legal privilege or by doctor-patient confidentiality, data that includes the content of individuals' non-public communications)?

No, it was crowd-sourced with paid workers who agreed to work on this task.

### N.2.12 Does the dataset contain data that, if viewed directly, might be offensive, insulting, threatening, or might otherwise cause anxiety?

No.

### N.2.13 Does the dataset relate to people?

Especially the Action-Genome-Edit data depicts people in their homes.

### N.2.14 Does the dataset identify any subpopulations (e.g., by age, gender)?

We do not know the exact recruitment for Action-Genome and Something Something videos (we build on top of these), but there are usually requirements such as speaking English.

### N.2.15 Is it possible to identify individuals (i.e., one or more natural persons), either directly or indirectly (i.e., in combination with other data) from the dataset?

If someone really tried, they might be able to identify some of the people in Action-Genome-Edit since they are shown fully and in their home.

### N.2.16 Does the dataset contain data that might be considered sensitive in any way (e.g., data that reveals racial or ethnic origins, sexual orientations, religious beliefs, political opinions or union memberships, or locations; financial or health data; biometric or genetic data; forms of government identification, such as social security numbers; criminal history)?

No.

### N.2.17 Any other comments?

None.

### N.3 Collection process

*The answers to questions here may provide information that allow others to reconstruct the dataset without access to it.*

### N.3.1 How was the data associated with each instance acquired?

For the sub-datasets MagicBrush, Action-Genome-Edit and Something-Something-Edit the prompts were written by humans and in the case of MagicBrush, the edited images were also produced in collaboration with an AI editing tool (DALL-E 2).

### N.3.2 What mechanisms or procedures were used to collect the data (e.g., hardware apparatus or sensor, manual human curation, software program, software API)?

For the data we collected ourselves with humans (Action-Genome-Edit), we used Amazon Mechanical Turk.

### N.3.3 Who was involved in the data collection process (e.g., students, crowdworkers, contractors) and how were they compensated (e.g., how much were crowdworkers paid)?

We worked with 7 workers from Amazon Mechanical Turk and paid them $0.22 USD per example.

### N.3.4 Over what timeframe was the data collected?

Around a week at the end of April 2024 for the main collection of Action-Genome.

### N.3.5 Were any ethical review processes conducted (e.g., by an institutional review board)?

No.

### N.3.6 Does the dataset relate to people?

Only in the sense that the prompts were written by people and that 1-2 dataset we build on top of depicts people.

### N.3.7 Did you collect the data from the individuals in question directly, or obtain it via third parties or other sources (e.g., websites)?

We collected it directly from AMT.

### N.3.8 Were the individuals in question notified about the data collection?

Workers on AMT see the posting with details like price and task description.

### N.3.9 Did the individuals in question consent to the collection and use of their data?

They implicitly agreed to various uses through the terms of service by MTurk.

### N.3.10 If consent was obtained, were the consenting individuals provided with a mechanism to revoke their consent in the future or for certain uses?

I am not sure about that part of MTurk's legal agreement.

### N.3.11 Has an analysis of the potential impact of the dataset and its use on data subjects (e.g., a data protection impact analysis) been conducted?

No.

### N.3.12 Any other comments?

None.

### N.4 Preprocessing/cleaning/labeling

*The questions in this section are intended to provide dataset consumers with the information they need to determine whether the "raw" data has been processed in ways that are compatible with their chosen tasks.*

#### N.4.1 Was any preprocessing/cleaning/labeling of the data done (e.g., discretization or bucketing, tokenization, part-of-speech tagging, SIFT feature extraction, removal of instances, processing of missing values)?

We mainly filtered out image pairs with too many changes: We told workers to discard images with too many (or in rare cases too few changes).

#### N.4.2 Was the "raw" data saved in addition to the preprocessed/cleaned/labeled data (e.g., to support unanticipated future uses)?

It was not directly saved but can be accessed again by downloading the original sources we build upon.

#### N.4.3 Is the software used to preprocess/clean/label the instances available?

We provide scripts on how to go from raw videos/frames to the cleaner ones on our repository.

#### N.4.4 Any other comments?

None.

### N.5 Uses

*These questions are intended to encourage dataset creators to reflect on the tasks for which the dataset should and should not be used. By explicitly highlighting these tasks, dataset creators can help dataset consumers to make informed decisions, thereby avoiding potential risks or harms.*

#### N.5.1 Has the dataset been used for any tasks already?

We used it to train an image editing model. We expect similar applications, also to video generation models.

#### N.5.2 Is there a repository that links to any or all papers or systems that use the dataset?

Our code repository, or [MagicBrush](https://github.com/OSU-NLP-Group/MagicBrush).

#### N.5.3 What (other) tasks could the dataset be used for?

Training models for video generation, change descriptions (i.e. Vision-and-Language LLMs) or discrimination of two similar images.

#### N.5.4 Is there anything about the composition of the dataset or the way it was collected and preprocessed/cleaned/labeled that might impact future uses?

Not that I can think of.

#### N.5.5 Are there tasks for which the dataset should not be used?

Unsure.

#### N.5.6 Any other comments?

None.

### N.6 Distribution

#### N.6.1 Will the dataset be distributed to third parties outside of the entity (e.g., company, institution, organization) on behalf of which the dataset was created?

We will fully open-source it and provide access via Zenodo/json files as well as Huggingface Datasets.

#### N.6.2 How will the dataset will be distributed (e.g., tarball on website, API, GitHub)?

Both Zenodo and Huggingface datasets.

#### N.6.3 When will the dataset be distributed?

The weeks after submission to NeurIPS Dataset & Benchmark track, so in June 2023.

#### N.6.4 Will the dataset be distributed under a copyright or other intellectual property (IP) license, and/or under applicable terms of use (ToU)?

We stick with the standard open-source license: MIT License

#### N.6.5 Have any third parties imposed IP-based or other restrictions on the data associated with the instances?

Something Something is the only dataset with a restricted license.

#### N.6.6 Do any export controls or other regulatory restrictions apply to the dataset or to individual instances?

[Official access and licensing](https://developer.qualcomm.com/software/ai-datasets/something-something) of Something Something dataset.

#### N.6.7 Any other comments?

None.

### N.7 Maintenance

*These questions are intended to encourage dataset creators to plan for dataset maintenance and communicate this plan with dataset consumers.*

#### N.7.1 Who is supporting/hosting/maintaining the dataset?

The main author is responsible for ensuring long-term accessibility, which relies on Zenodo and Huggingface.

#### N.7.2 How can the owner/curator/manager of the dataset be contacted (e.g., email address)?

benno.krojer@mila.quebec (or after I finish my PhD benno.krojer@gmail.com)

#### N.7.3 Is there an erratum?

Not yet!

#### N.7.4 Will the dataset be updated (e.g., to correct labeling errors, add new instances, delete instances)?

Not sure yet. If we find that people are interested in the data or trained model, we will continue our efforts.

**N.7.5** **If the dataset relates to people, are there applicable limits on the retention of the data associated with the instances (e.g., were individuals in question told that their data would be retained for a fixed period of time and then deleted)?**

No.

**N.7.6** **Will older versions of the dataset continue to be supported/hosted/maintained?**

If there ever was an update, yes.

**N.7.7** **If others want to extend/augment/build on/contribute to the dataset, is there a mechanism for them to do so?**

Since we use a non-restricting license (MIT license), anyone can build on top or include in their training data mixture.

## N.8  Any other comments?

No. We hope the data is useful to people!

