# OpenReview forum: "Learning Action and Reasoning-Centric Image Editing from Videos and Simulation"
_NeurIPS.cc/2024/Datasets_and_Benchmarks_Track — NeurIPS 2024 Track Datasets and Benchmarks Spotlight_

### Official Review · Reviewer_yzTK · 2024-07-15

**Rating:** 8
**Confidence:** 4
**Correctness:** Yes, the dataset is sound and clearly…
**Clarity:** Yes. Supporting materials strengthen …

**Review:**

The introduced dataset is well-motivated and results in impressive performance gains on the important task of image editing. Further, this work produces a benchmark to evaluate the task more broadly than before, and a metric to evaluate more fairly than before. The Supplemental materials enrich the overall results of the paper. For more detail, please see Strengths.

Thanks to the authors for their thorough rebuttal and the reviewers for their thoughtful comments. After taking these into account, I want to reiterate I agree with the other reviewers that this is a strong paper. It is a clear accept and should be considered for spotlight / oral.

**Strengths:**

The dataset yields strong performance across a variety of editing tasks. It along with the introduced evaluation metric are well-motivated with qualitative and quantitative results
- Humans stated the method was 1.5x-6x more successful than the previous SOTA at several tasks testing actions, human-object-interaction and reasoning. In tasks previous SOTA MagicBrush was better, the dropoff was not massive.
- Performance on introduced metric based on DiscEdit is also strong, being selected more often than MagicBrush across several tasks, often by >10%. This metric is motivated by the nonsensical results from previous reported metrics L1, DINO, CLIP-I on this task.
- Handles diverse tasks such as articulation (e.g. human pick up glass), movement (e.g. rotate or insert object), reasoning (e.g. put cloth on water bottle)
- The lack of “minimal image pairs” is identified as the cause of prior model failings. Qualitative examples highlight this as well as that introduced data has minimal pairs, which results in expected edits.
- Additional results across the Supplemental PDF, data, code, and demo enrich findings from the main paper.

**Additional Feedback:**

- L83 referrring should be referring
- The model used should be clarified. Although I believe it can be inferred the model used for fine-tuning is MagicBrush, I couldn't find this explicitly stated anywhere.

**Documentation:**

Yes, Section A of Supplement contains all needed related info.

**Ethics:**

No.

**Limitations:**

Yes.

**Opportunities For Improvement:**

Table 2 is slightly unfair to prior work without context
- While AURORA trains on a variety of datasets relevant to each evaluated task, prior work is not trained on all of these. For example, MagicBrush is not trained on Kubric, so it is probably not surprising it is significantly worse than AURORA on this task. Although Table 1 makes it clear MagicBrush doesn’t handle the skills of “action” or “reasoning,” reading Table 2 standalone this would not be clear. Perhaps an asterisk could be used, or some other option, to set this up.

**Relation To Prior Work:**

Yes, this is clearly analyzed throughout the paper.

**Summary And Contributions:**

This paper approaches the task of text-conditional image editing. It focuses on general image editing: modifying object attributes, style, location, and applying actions to the object. To this end, it presents a training and evaluation dataset as well as a new evaluation metric for this setting. It shows shortcomings of previous metrics and shows fine-tuning prior work significantly improves performance on its evaluation set and metric.

---

> ### Author Rebuttal · Authors · 2024-08-14
>
> Hello,
>
> Thank you for the positive feedback as well as the constructive criticisms! Your write-up clearly shows you properly engaged with our work and paid attention to details such as the exact interpretation of the empirical results.
> We are glad that you found both our **dataset and metric well-motivated** and acknowledged that ***“the method was 1.5x-6x more successful than the previous SOTA at several tasks testing”*** (according to humans)!
> It is also nice to read that you found our contribution beyond the main text useful and well-documented such as supplemental, code etc.
>
> Your **main feedback for improvement was about a fair comparison of past methods** in Table 2:
> As you noted, past models are not trained on some of the datasets and thus it is not surprising they perform lower on those.
> First, we like your suggestion of adding an asterisk and explain the comparison!
> We agree it is not surprising we outperform e.g. MagicBrush across all 8 tasks. On the other hand, we did find it is hard to learn from videos and actions, much harder than the more established object replacement or attribute changes. In earlier experiments, the model would often become more noisy/unreliable or performance would drop on anything but the IID training data (e.g. on MagicBrush) if the data quality is low or the ratio of datasets is off.
> For that reason we wanted to show how far you can get with adding this additional new quality data, and that performance on MagicBrush does not drop at the same time, plus aspects like transfer to zero-shot WhatsUp.
>
> Finally, it is a good catch that we did not specify the exact training setup (initial checkpoint, mix of data) in the main paper, only in the appendix. We will definitely add this with the additional page for camera ready.
>
> **Thank you for pointing these out and we hope this addressed all your concerns!**

---

### Official Review · Reviewer_TXfY · 2024-07-23
**Exciting new dataset and results that pushes the task forward**

**Rating:** 9
**Confidence:** 4

**Review:**

i believe this paper to be strong with a new dataset that has been collected for this task with a nice amount of detail and motivation behind its collection and for its collection. The paper is generally easy to understand as well. Experiments are well presented and include a nice amount of thought behind their choice as well as including good discussion. The paper could have some aspects clarified including whether the collection process had approval from an IRB as this is mentioned as N/A in the checklist as well as for the human participant judgement.

**Strengths:**

The paper presents a challenging and exciting new dataset for the task of reasoning centric image editing that has been collected from a variety of different datasets and annotated using human annotators.

The experiments are highly detailed and showcase why the dataset was necessary but also challenges previous metrics for the area. A particular personal highlight are the results that showcase that the naive copy can outperform SotA methods on this task when older metrics are considered.

**Additional Feedback:**

Epic Kitchen -> Epic Kitchens
Line 158: lenghty -> length
Line 277 minmal -> minimal
Line 288 The quotation marks around stabilizes are flipped to what they should be.

As a final note, I really enjoyed seeing and reading the Behind the Scenes section in the appendix (Appendix L) I think it is something that can add value to the project and the paper to give a widening context, if only to let others know the directions which were found to be dead ends. It's something that I wished was more commonplace (if only I could convince others to do it!)

**Clarity:**

The paper was generally easy to read and understand. The layout was sensible and included a lot of the necessary detail. A minor comment to improve here would be the number of times that the appendix was referenced within the main paper. This tended to break the flow as experimental results or figures were referenced in the main paper but not viewable.

**Correctness:**

The dataset construction is good and well motivated, I particularly like how the dataset was chosen to be annotated and collected from a variety of different datasets which allow for interesting axes of experimentation.

**Documentation:**

The datasheets for datasets has been attached as an appendix and the required questions from the NeurIPS submission have been filled out satisfactorily within the appendix. The URL and code are also provided.

**Ethics:**

The ethics of the study appears to be well thought out, reasoned, and implemented. The only question I would have for the authors at this point was whether this study was reviewed by the University's review board (IRB) or similar.

The documentation included the amount that participants were being paid, as well as showcasing that it was more than US minimum wage and included some evidence of happy participants. There is also a nice ethical snd societal impact discussion within the

**Limitations:**

Yes, the authors discussed the limitations within the supplementary material and I believe that these are sufficient.

**Opportunities For Improvement:**

The paper as it currently stands is strong and I don't believe includes many areas for improvement, or at least only minor changes/questions arise from reading the paper. These are:
1. It would be good to have more definition/discussion behind the human judgement of 0-100 scale. Why was this chosen beyond a 1-10 scale? How were participants calibrated to these scores?
2. It is not clear how the USD per hour that the participants achieved was calculated, do you have the average time to complete a caption?
3. Was the collection process ratified by the university ethics board or similar?
4. The paper often refers to the appendices which can take the reader out of the section, especially as results/figures are discussed but not present within the main paper.
5. The paper includes some spelling, grammar, and typesetting mistakes (quotation marks) which could be improved to help with flow of reading.

**Relation To Prior Work:**

The paper clearly distinguishes how the paper relates to prior work, including many similar datasets and/or methods which have tried to include image editing and goes into a lot of detail with experiments as to why these are unsuitable. From my point of view there are no missing published related works that I could think of as relevant that were not included within the work.

**Summary And Contributions:**

The paper proposes a new dataset for reasoning centric image editing and collects this from a number of current datasets with a large variety between sim/real, video/image, and task/domains. These have been collected with AMT workers (paid well) and results showcase how current methods struggle on the new benchmark. Of particular note are the experiments and qualitative results which go into detail regarding issues with current methods and evaluation metrics.

---

> ### Author Rebuttal · Authors · 2024-08-14
>
> Thank you for your positive review, and especially a **big thank you for engaging with the work in so much detail**, such as caring about things beyond the main table/results/method. We put care into doing science that is easy to reproduce, understand and where human evaluation/crowdsourcing is done well, and it is clear you acknowledged those parts of the paper that usually go under the radar with many reviewers!
>
> Specifically, you pointed out: *“The experiments are highly detailed and showcase why the dataset was necessary but also challenges previous metrics for the area. A particular personal highlight are the results that showcase that the naive copy can outperform SotA methods on this task when older metrics are considered.”*
> We are glad you found the paper *“easy to understand”* (we went through many revisions of how to frame it best) and that there was a *“nice amount of thought behind choosing our experiments”*.
>
> We also thank you for the detailed and truly helpful comments that show us where some more clarifications are needed. Let’s discuss each one briefly:
>
> 1. ***"More discussion behind the human judgement of 0-100 scale”***: We give humans three options regarding “Did the output adhere to the edit prompt semantically?”: 1) no, 2) partially, 3) yes. And show in more detail in appendix D2 how a judge should choose between the three (e.g. not focusing on aesthetics). We choose the 0-100 scale simply for presentation purposes but acknowledge that it can be misleading! We will clarify in the respective table and maybe emphasize it more in the text itself (already briefly mentioned in line 236).
> 2. ***“Not clear how the USD per hour was calculated”***: The honest answer is that it was not directly measured but came from two aspects: A) the first author estimating that it would take 20-40 seconds per caption based on themselves doing it and then adding some seconds if someone takes longer. This would be 90 to 180 captions per minute and thus 20 to 40 dollars per hour. Even with some delays navigating between individual AMT examples and smaller work breaks, this is still a good pay of more than 15 dollars per hour. B) Additionally, we were in close contact with the final pool of workers and received positive feedback that the pay was good. In earlier test rounds we had paid a bit less and had promised to pay more once we feel everyone has understood the task well and we can scale it up.
> 3. **Regarding the paper formatting, typos**, we will revise the paper for camera ready! Regarding “the paper often refers to the appendices which can take the reader out of the section”, we agree and were somewhat aware during writing. As another reviewer made a similar remark, we will make sure to move some details into the main paper with the additional page and make sure the appendix references break the flow less.
> 4. You also asked ***“whether this study was reviewed by the University's review board (IRB) or similar”***. The short answer is no. For these types of datasets (i.e. annotating already existing images with short non-personal captions) it is very common in ML research to not seek IRB approval. But additionally we did look into our official university guidelines: (https://www.mcgill.ca/research/files/research/policy_on_the_ethical_conduct_of_research_involving_human_participants.pdf) and concluded that: our institutional policy does not require REB/IRB approval for knowledge that is in public domain and is not subjective.
>
> P.S.: It made my day when the reviews came out that someone acknowledged the “Behind the Scenes” section. Since the start of my PhD I always wanted to write papers a bit more accessible and honest, easier for someone new to the field to know what’s really going on behind all the big claims everyone is making. So your comment is **very encouraging and means I will keep writing papers with more of this transparent-science angle** in the future!

---

### Official Review · Reviewer_XmHR · 2024-07-24
**AURORA Review**

**Rating:** 8
**Confidence:** 4
**Correctness:** Yes!
**Clarity:** Yes!

**Review:**

This dataset is significant for a large number of researchers focused on the ability of generative models to perform action and reasoning centric edits to images.

The motivation is salient and the contribution is of high quality, with a new metric for assessing edits that is needed in image editing and in-painting.

The pros are:
- annotated many action/reasoning prompts for image pairs across a variety of data sources
- include negative/no change prompts for these images too (I think? needed for their metric)
- data is already released onto hugging face

The cons are:
- could have included more data and prompt types
- some edits like "make the yellow pot fall over" actually flip the pot upside down, this isn't falling over but flipping. Means that the granularity / accuracy of the change is called into question. Means that prompts will be more coarse than they ideally should be.

**Strengths:**

The image editing community lacks metrics grounded in prompts, this paper provides a new one, DiscEdit. It is a sensible metric too.

Anyone who is trying to understand whether models they have trained are able to do visual reasoning can now use this this benchmark to understand their editing ability, or even their captioning ability.

Generative world simulators have lacked good means of assessing and fine-tuning on actions. Next token/autoregressive/future prediction has visual quality metrics but methods like Video Language Planning that generate prompts for describing hypothetical future states and then produce video to match these prompts can lack ways of assessing how well the video fits the prompt. AURORA provides a way to understand how well generations match.

Action and Reasoning -centric generation is probably the next evolution of generative models now that image quality and adherence to initial prompts is well covered by existing models. This benchmark and dataset and new DiscEdit metric show up at a great time to be of use to this community.

**Additional Feedback:**

N/A

**Documentation:**

Yes! I can access the data easily and it is clear.

**Ethics:**

No.

**Limitations:**

A remaining limitation is an understanding of the landscape of actions and reasoning that can be considered. For example, there are many many verbs that one could use in a reasoning context -- even visual genome has a large number of verbs. It could do to included more analyses of the 400 edit prompts that were used and the type of coverage this has compared to what exists in realistic data.

**Opportunities For Improvement:**

The paper is limited in details in the main body, as it contributes a lot and page limits are difficult. I think that the main body of the paper could be better distributed, with some shorter sections and a few more interesting results from the 29 page supplement included.

**Relation To Prior Work:**

Image inpainting edits have been a topic of recent interest and the related work section does a good job of surveying what's been done so far and then comparing against this body of work. An existing direction that could be further explored is the relation to VLMs and their capabilities.

**Summary And Contributions:**

This paper introduces a dataset and benchmark for image editing that goes beyond simplistic edits and includes descriptive, semantic edits. An example is to change an image from a person typing on a computer (an input image), to a person drinking a cup of water with their right hand (a prompt).

This dataset is focused on action-centric and reasoning-centric instructions for tuning. It includes a benchmark for assessing edit types.

Further, the authors include a new metric for assessing image edits, DiscEdit. This metric considers that most image editing metrics are bad, and reward just copying the input image as better than actually doing something. Specifically, this metric requires the additional annotation of a prompt that should not change the image. However, this additional annotation leads to a new, better way of evaluation that is interpretable and promising.

Finally this paper introduces a model trained on the dataset that is state-of-the-art in preforming image edits.

---

> ### Author Rebuttal · Authors · 2024-08-14
>
> Hello,
>
> We appreciate the time you took to engage deeply with our work and the overall positive conclusion! We were especially happy to read some of your more higher-level positive comments such as *“Action and Reasoning-centric generation is probably the next evolution of generative models”* and that *“this benchmark and dataset and new DiscEdit metric show up at a great time to be of use to this community”*. In more detail, we are glad DiscEdit seems like a good and “sensible” metric to you, that we provided a lot of annotations (both from crowdworkers, and yes the authors also annotated a lot!) and that everything is easily accessible (Huggingface etc.).
> For the rest of the response, we want to **focus on smaller feedback of yours first and then discuss in more length the one feedback you gave on exploring the distribution of verbs/actions** covered in AURORA:
>
> *“Include more data and prompt types”*: Can you clarify what you think would be nice to have in the dataset? We are definitely exploring future directions for this project, so  we are curious to hear what you would like to see. As described in section L in the Appendix (Behind the scenes), we were initially looking into more larger scale diverse data from the wild but found it simply too noisy. With more elaborate filters and automatic pipelining, it might be possible!
>
> *“Some edits like "make the yellow pot fall over" actually flip the pot upside down, this isn't falling over but flipping. Means that the granularity / accuracy of the change is called into question”*:
> We genuinely thank you for finding this issue in our Fig. 2, we fixed it! This is not the case in the actual data and only an issue in the figure (we checked the json again to make sure). It probably happened since we were manually trying to make the figure fit well and look nice with collaborators (e.g. it was originally in the data “yellow flower pot” but I shortened it).
> There is definitely some noise in the natural datasets but the simulation one is quite clean.
>
> *“limited in details in the main body”*: We agree and hope to move more details to the main paper for the camera ready (and later Arxiv versions).
>
> *Related VLM capabilities*: Similarly, this is a good suggestion. We hope to have a smaller section on broader related work discussing different approaches to “minimal change” understanding in VL, given the additional page.
>
> *“The remaining limitation is an understanding of the landscape of actions and reasoning that can be considered. For example, there are many many verbs that one could use in a reasoning context -- even visual genome has a large number of verbs”*:
> It would be fascinating to see if there is a different distribution of verbs between broad generic VL captioning datasets and datasets tailored for action-editing! Even before conducting further experiments, we can already say that a lot of edits have the generic verbs like “move”, which is nonetheless often still a complex task: “Move the cup closer to the plate” is a very different move than “Move the hand closer to their hair”, where the exact action required is implicit in the scene/affordances/angles and not reflected in the textual “move”. I think this is somewhat inherent to the editing task itself where captions tend to be shorter than traditional caption datasets, often with simpler verbs “make OBJECT ATTRIBUTE” (make the horse darker) or “replace/add OBJECT”. So another interesting comparison would be the distribution of verbs in traditional (object/attribute) editing vs. our focus on action editing.
> Finally, we note that a lot of complexity in our data comes from other linguistic constituents such as prepositions or adjectives/adverbs, e.g. “Move the hand slightly closer under the table with the finger pointing upward” where {slightly, closer, under, upward} are all interesting to understand but not verbs.
> **To study the frequency differences to established caption datasets, we visualize the verb and noun distribution in MS-COCO, AURORA as well as the four subsets of AURORA. See PDF figure caption for the details!**
> Overall, the verbs are less diverse but  as described above a lot of the complexity comes from other textual or visual aspects. On top, the verbs are quite different to COCO and notably also quite different to more established object/attribute-centric editing.
> Also note that while “make” is a very frequent verb, it can often be accompanied with one of the other verbs like “make the person stand up”.

---

### Official Review · Reviewer_W52U · 2024-08-04
**A data set for image editing**

**Rating:** 7
**Confidence:** 3
**Correctness:** The work appears correct.

**Review:**

Overall, I found the author's case compelling for why a better data set is needed.  My primary concerns are with the utility of such a data set more generally (see limitations section below).  The repository is well-document and the work was clearly written.

**Strengths:**

The data set can easily be adopted by current practitioners.

The authors demonstrated empirically that their data set leads to performance improvements.

**Additional Feedback:**

n/a

**Clarity:**

The paper is clearly written and well-motivated.  There were a few minor typos here and there (all put commas after "e.g."!), but none that impacted my understanding significantly.

**Documentation:**

The GitHub repo was well-documented.  It was clear how to use the data set.

**Ethics:**

None.

**Limitations:**

Only minor limitations were discussed in the main text.  The main limitation for me is that, while this data set may provide a better training platform than existing data sets, it isn't clear that "reasoning" can be attained only with data (or at least the amount of data being considered here).  The metrics are also a bit of a limitation in these applications.  While the authors do propose an alternative to existing metrics, which themselves aren't really all that great, even their proposed metric falls short of human evaluators.

For me, the above concerns could potentially limit the longevity and utility of this data set.

**Opportunities For Improvement:**

Details about how the human evaluators assessed the outputs is missing in the write-up.  It seems straightforward here, but it should be included.

**Relation To Prior Work:**

Related work is covered appropriately.

**Summary And Contributions:**

The authors identify issues and the propose solutions for training and evaluating models for image editing tasks.  Their primary contribution is a carefully curated data set of minimal changes that is much less noisy then existing data sets.  They also propose an interesting metric that attempts to encode the simple knowledge that tasks that require small changes should result in fewer changes to an image than global edits.

---

> ### Author Rebuttal · Authors · 2024-08-14
>
> Thank you for engaging critically with our work and identifying both positives as well as some potential limitations! We appreciate that you found our paper “well-motivated” (“author's case compelling for why a better data set”) and “clearly written”, and  the dataset/code easy to adopt & well documented. Finally, you also acknowledged that we show the empirical value of the data.
> We will first briefly address your minor negative feedback (*documenting the human evaluator setup*), and then discuss your major concern (*“utility of such a data set more generally”*) in more depth.
>
> In the main paper (Section 5.2) we write:
> “We ask humans to rate the absolute edit success (0=none, 50=partial, 100=full) as well as comparison (i.e. win-rates) between different models. We focus on the former in our main results as it allows us
> to compare task difficulty. We ensure that evaluators (we pick the best three evaluators from crowd-sourcing AURORA) pay most attention to the correct (semantic) interpretation of the edit prompts [...]”. We also provide more details in the appendix (D.2) such as screenshots of how we instructed workers and the interface itself. In short, we tell them to focus mostly on the semantics (“Is the meaning of the edit prompt adhered to?”) and not aesthetics of the generated edit. We found it more productive and helpful to provide several examples initially, get the worker’s judgment, then provide feedback; compared to describing it abstractly.
>
> **The main concern is that *“[...] it isn't clear that "reasoning" can be attained only with data (or at least the amount of data being considered here)”***. First, we do agree that the amount of data (or diversity) needs to be scaled up even more for robust editing models. As of now it is still not robust enough. We initially experimented with YouTube videos in the wild (which is more scalable than curated data) but found that it does not work as well (e.g. see appendix L). We also experimented with using GPT-V instead of human annotators, but again we favored quality over quantity. However with a lot of prompting and filtering a GPT-V and YouTube video approach could be viable! And we might explore it in future work.
> Lastly, we also partially agree that reasoning can’t come from data alone. But even for systems with strong potential to reason or build an internal world model, they have to learn it from somewhere. Human infants also receive an interesting (arguably curated from parents) environment to learn a lot of skills. Generally, at this stage of AI research good quality data often seems to be the biggest lever - but we hope that modeling and training strategy can advance in parallel and make use of datasets like AURORA. While datasets naturally run out of fashion after many years (only very few are still used from 5+ years ago), we hope that AURORA due to its high quality and easy accessibility will be used for the next few years.
>
> Again, thank you for this fruitful discussion that led us to reflect on the (long-term) value of our work!

---

### Author Rebuttal · Authors · 2024-08-14

We want to thank all the reviewers for engaging rigorously and constructively with our work!

The review process can sometimes be chaotic and difficult but with all four reviewers we really felt everyone had properly read, understood and fairly assessed the pros and cons of our paper.

**Reviewers identified a lot of positives:**

1. All four reviewers agreed the paper (and its dataset/evaluation) is well-motivated and easy to follow - overall a strong contribution: *“case compelling for why a better data set is needed”* (W52U), *“motivation is salient and the contribution is of high quality, with a new metric [...]”* (XmHR), *“The experiments are highly detailed and showcase why the dataset was necessary but also challenges previous metrics for the area”* (TXfY), *“dataset is well-motivated and results in impressive performance gains”* (yzTK), *“This benchmark and dataset and new DiscEdit metric show up at a great time to be of use to this community”* (XmHR)
2. In more detail, our DiscEdit metric and a critical revisit of established metrics was well-received: *“The image editing community lacks metrics grounded in prompts, this paper provides a new one, DiscEdit. It is a sensible metric too.”* (XmHR), *“Of particular note are the experiments and qualitative results which go into detail regarding issues with current methods and evaluation metrics”* (TXfY), *“A particular personal highlight are the results that showcase that the naive copy can outperform SotA methods on this task when older metrics are considered.”* (TXfY)
3. Our rigor in curating the data and releasing it was also appreciated: *“easily be adopted by current practitioners”* (W52U), *“data is already released onto hugging face”* (XmHR), *“how the dataset was chosen to be annotated and collected from a variety of different datasets which allow for interesting axes of experimentation.”* (TXfY)
4. Our results were thought to be convincing: *“Experiments are well presented and include a nice amount of thought behind their choice”* (TXfY), *“Humans stated the method was 1.5x-6x more successful than the previous SOTA at several tasks testing actions, human-object-interaction and reasoning”*, (yzTK) *“demonstrated empirically that their data set leads to performance improvements”* (W52U)
5. Finally, our supplemental material was detailed and appreciated for its honest science communication: *“As a final note, I really enjoyed seeing and reading the Behind the Scenes section in the appendix (Appendix L) I think it is something that can add value to the project and the paper to give a widening context, if only to let others know the directions which were found to be dead ends. It's something that I wished was more commonplace (if only I could convince others to do it!)”* (TXfY), *“Supplemental materials enrich the overall results of the paper”* (yzTK)


We also received several constructive comments and feedback. While we could not spot a major theme throughout the four reviews wrt. to shortcomings, we will still **summarize the most important questions/limitations, and how we addressed them**:
1. We used our 1-page PDF to show plots and analysis as a response to XmHR: *“The remaining limitation is an understanding of the landscape of actions and reasoning that can be considered. For example, there are many many verbs that one could use in a reasoning context -- even visual genome has a large number of verbs”*. We will include this analysis in the camera ready version and find it a very insightful analysis, thank you XmHR!
2. We clarified the details of how human were instructed to rate and compare model outputs to reviewer W52U and TXfY
2. Two reviewers suggested to have more details in the appendix (XmHR) and avoid breaking the flow with appendix references (TXfY)
4. Reviewer yzTK pointed out that the way we present the main results in Table 2 is not fully fair and requires some more clarification in the figure caption and with an asterisk, which we will adopt in the final paper.
5. Reviewer TXfY had questions about crowdsourcing details, specifically IRB approval and how the worker pay was decided, which we hopefully addressed: 1) our university guidelines do not require IRB for this kind of collection, 2) we estimated the duration to complete an example and communicated with workers to make sure they feel well compensated.
6. Finally, we also discussed the long-term utility and validity of AURORA with reviewer W52U, i.e. is data alone enough to learn reasoning?

The additional PDF contains additional analyses of verb/noun distributions in a common captioning dataset (MS-COCO) vs. our editing dataset. This was inspired by an interesting suggestion from reviewer XmHR. See the response to XmHR for a detailed interpretation of the plot.

**Again, we thank all the reviewers and the AC for this constructive process so far and are looking forward to engaging more in the following weeks!**

---

### Decision · Program_Chairs · 2024-09-26

**Decision:**

Accept (Spotlight)

**Comment:**

The reviewers unanimously found the paper to be of high quality and significance for acceptance.